# SoK: A Generalized Multi-Leader State Machine Replication Tutorial

Michael Whittaker
*UC Berkeley*

Neil Giridharan
*UC Berkeley*

Adriana Szekeres
*University of Washington*

Joseph M. Hellerstein
*UC Berkeley*

Ion Stoica
*UC Berkeley*

## Abstract

MultiPaxos and Raft are the two most popular and widely deployed state machine replication protocols. There is a more sophisticated family of generalized multi-leader state machine replication protocols like EPaxos, Caesar, and Atlas that have better performance, but they are extremely complicated and hard to understand. Due to their complexity, they have seen little to no industry adoption, and academically there has been a lack of clarity in analyzing, comparing, and extending the protocols. This paper is a tutorial on generalized multi-leader protocols. We explain why the protocols work the way they do, what they have in common, where they differ, which parts of the protocols are straightforward, which are more subtle than they appear, and so on. In doing so, we present four new generalized multi-leader protocols, identify key insights into existing protocols, and taxonomize the space.

## 1 Introduction

State machine replication protocols are a critical component of many fault tolerant distributed systems [4, 5, 8, 27, 29]. Given an arbitrary deterministic state machine, like a key-value store or a relational database, a state machine replication protocol can be used to deploy multiple copies, or replicas, of the state machine while guaranteeing that the states of the replicas stay in sync and do not diverge.

The most popular and widely deployed state machine replication protocols are Paxos [5, 8, 13] and Raft [1, 2, 23, 27]. These protocols have two distinguishing characteristics. First, they are leader based. All communication is funneled through a single leader. Second, these protocols totally order state machine commands into a log and have state machine replicas execute the commands in log order. Every replica executes the exact same commands in the exact same order.

There is another family of *generalized multi-leader* state machine replication protocols—including EPaxos [22], Caesar [3], and Atlas [10]—that improve the performance of protocols like MultiPaxos and Raft along these two dimensions. These protocols are **multi-leader** and avoid being throughput bottlenecked by a single leader. They are also **generalized** [14, 19]. This means that the protocols are based on dependency graphs. Every replica executes non-commuting commands in the exact same order, but the replicas are free to execute commuting commands in any order. As a result, commuting commands do not interfere with one another.

Unfortunately, these generalized multi-leader protocols are extremely complicated. Paxos has a well known reputation for being complex [16, 23, 30], and these generalized multi-leader protocols are significantly more complex than that. They require a strong understanding of more sophisticated Paxos variants like Fast Paxos [15] and are overall less intuitive and more nuanced. It's hard to measure this complexity precisely, but there are indications that the protocols are complicated. EPaxos, for example, had several bugs go undiscovered for years despite the popularity of the protocol [26]. Through personal conversations, we have also found that even domain experts find these protocols challenging to fully understand.

This complexity has negative consequences in industry and academia. The performance advantages of generalized multi-leader protocols make them an attractive option for industry practitioners. Despite this, generalized multi-leader protocols have little to no industry adoption. We postulate that this is largely due to their complexity. The engineers in [6] explain that implementing a state machine replication protocol requires making many small changes to the protocol to match the environment in which it is deployed. Making these changes without a strong understanding of the protocol is infeasible. Academically, it is challenging to compare and contrast the various protocols. They all seem very similar, yet vaguely distinct. This also makes it difficult to extend the protocols with further innovations. There are dozens of state machine replication protocols in the literature, yet relatively few generalized multi-leader variants.

This paper is a tutorial on generalized multi-leader state machine replication protocols. Our goal is to answer questions such as: What problem do these protocols address? How can I choose between the various protocols? Why do these protocols work the way they do? What do they have in common? Where do they differ? Which parts of the protocols are straightforward? Which are more subtle than they appear? Are there simpler variants out there? What trade-offs do the protocols make, and which points in the design space are still unexplored?

The tutorial has four parts, and in each part, we introduce a new protocol. First, we present the simplest possible generalized multi-leader protocol, which we called **Simple BPaxos** (Section 4). Simple BPaxos sacrifices performance for simplicity and is designed with the sole goal of being easy to understand. Simple BPaxos is the kernel from which all other generalized multi-leader protocols can be constructed. It en-

capsulates all the mechanisms and invariants that are common to the other protocols.

Second, we introduce a purely pedagogical protocol called **Fast BPaxos** (Section 6). Fast BPaxos achieves higher performance than Simple BPaxos, but it is unsafe. The protocol does not properly implement state machine replication. Why study a broken protocol? Because understanding why Fast BPaxos does *not* work leads to a fundamental insight on why other protocols do. Specifically, we discover that generalized multi-leader protocols encounter a fundamental tension between agreeing on commands and ordering commands. The way in which a protocol handles this tension is its key distinguishing feature. We taxonomize the protocols into those that avoid the tension and those that resolve the tension.

Third, we introduce **Unanimous BPaxos**, a simple *tension avoiding* protocol (Section 7). We describe how tension avoiding protocols carefully enlarge quorum sizes to sidestep the tension. We also explain how Basic EPaxos [22] and Atlas [10] can be expressed as optimized variants of Unanimous BPaxos.

Fourth, we introduce **Majority Commit BPaxos**, a *tension resolving* protocol (Section 8). We describe how tension resolving protocols perform detective work to resolve the tension without enlarging quorum sizes. We also discuss the relationship between Majority Commit BPaxos and other tension resolving protocols like EPaxos [21] and Caesar [3].

In summary, we make the following contributions.

- We explain generalized multi-leader protocols carefully and thoroughly, bringing clarity to an otherwise dense area of popular research.

- We present four new generalized multi-leader state machine replication protocols: Simple BPaxos, Fast BPaxos, Unanimous BPaxos, and Majority Commit BPaxos.

- We identify a fundamental tension between agreeing on commands and ordering commands and use this insight to taxonomize generalized multi-leader protocols into those that avoid the tension and those that resolve it.

## 2   A Primer on State Machine Replication

Throughout the paper, we assume a system model in which messages can be arbitrarily dropped, delayed, and reordered. We assume machines can fail by crashing but do not act maliciously; i.e., we do not consider Byzantine failures. We assume that machines operate at arbitrary speeds, and we do not assume clock synchronization. Every protocol discussed in this paper assumes that at most $f$ machines will fail for some configurable $f$. If more than $f$ machines fail, the protocols remain safe, but won't be live.

### 2.1   State Machine Replication

**State machine replication** is the act of choosing a sequence (a.k.a. log) of values. A state machine replication protocol manages a number of copies, or **replicas**, of a deterministic state machine. Over time, the protocol constructs a growing log of state machine commands, and replicas execute the commands in log order. By beginning in the same initial state, and by executing the exact same commands in the exact same order, all of the state machine replicas are kept in sync. This is illustrated in Figure 1.

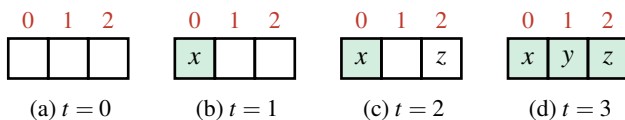

Figure 1: At time $t = 0$, no state machine commands are chosen. At time $t = 1$ command $x$ is chosen in slot 0. At times $t = 2$ and $t = 3$, commands $z$ and $y$ are chosen in slots 2 and 1. Executed commands are shaded green. Note that all state machines execute the commands $x, y, z$ in log order.

State machine replication builds on the simpler problem of **consensus**. Rather than choosing a sequence of values, consensus involves choosing a single value. State machine replication protocols like MultiPaxos implement state machine replication using one instance of consensus for every log entry, so to understand state machine replication, we must first understand consensus. We review Paxos, the most popular consensus algorithm, and then extend Paxos to MultiPaxos.

### 2.2   Paxos

A Paxos [13] deployment that tolerates $f$ faults consists of an arbitrary number of clients, $f + 1$ nodes called **proposers**, and $2f + 1$ nodes called **acceptors**, as illustrated in Figure 2. To reach consensus on a value, an execution of Paxos is divided into a number of integer valued rounds (also known as ballots, epochs, terms, views, etc. [12]). Every round has two phases, Phase 1 and Phase 2, and every round is orchestrated by a single pre-determined proposer. If a proposer is responsible for executing a round, we sometimes say the proposer is the leader of the round.

When a proposer executes a round, say round $i$, it attempts to get some value $v$ "chosen" in that round. We'll define formally what it means for a value to be chosen momentarily. Paxos is a consensus protocol, so it must only choose a single value. Thus, Paxos must ensure that if a value $v$ is chosen in round $i$, then no other value besides $v$ can ever be chosen in any round less than $i$. This is the purpose of Paxos' two phases. In Phase 1 of round $i$, the proposer contacts the acceptors to (a) learn of any value that may have already been chosen in any round less than $i$ and (b) prevent any new values from being chosen in any round less than $i$. In Phase 2, the proposer

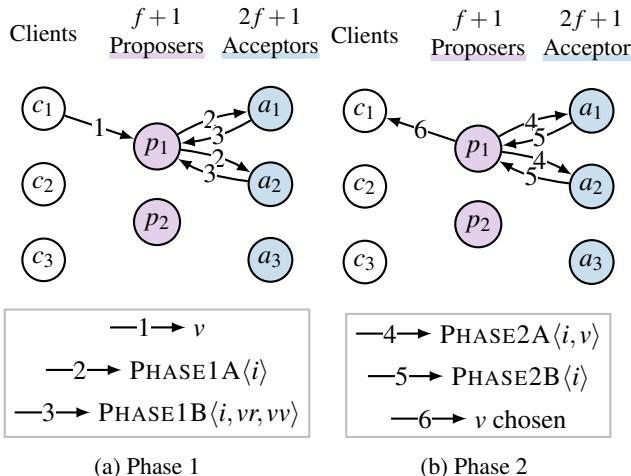

(a) Phase 1

(b) Phase 2

Figure 2: An example execution of Paxos ($f = 1$).

proposes a value to the acceptors, and the acceptors vote on whether or not to choose it. In Phase 2, the proposer is careful to only propose a value $v$ if it learned through Phase 1 that no other value has been or will be chosen in a previous round.

More concretely, Paxos executes as follows. When a client wants to propose a value $v$, it sends $v$ to a proposer $p$. Upon receiving $v$, $p$ begins executing one round of Paxos, say round $i$. First, it executes Phase 1. It sends PHASE1A$\langle i \rangle$ messages to the acceptors. An acceptor ignores a PHASE1A$\langle i \rangle$ message if it has already received a message in a larger round. Otherwise, it replies with a PHASE1B$\langle i, vr, vv \rangle$ message containing the largest round $vr$ in which the acceptor voted and the value it voted for, $vv$. If the acceptor hasn't voted yet, then $vr = -1$ and $vv = $ null. When the proposer receives PHASE1B messages from a majority of the acceptors, Phase 1 ends and Phase 2 begins.

At the start of Phase 2, the proposer uses the PHASE1B messages that it received in Phase 1 to select a value $v$ such that no value other than $v$ has been or will be chosen in any round less than $i$. Specifically $v$ is the vote value associated with the largest received vote round, or any value if no acceptor had voted (see [16] for details). Then, the proposer sends PHASE2A$\langle i, v \rangle$ messages to the acceptors. An acceptor ignores a PHASE2A$\langle i, v \rangle$ message if it has already received a message in a larger round. Otherwise, it votes for $v$ and sends back a PHASE2B$\langle i \rangle$ message to the proposer. If a majority of acceptors vote for the value (i.e. if the proposer receives PHASE2B$\langle i \rangle$ messages from a majority of the acceptors), then the value is chosen—this is the formal definition of when a value is chosen—and the proposer informs the client. This execution is illustrated in Figure 2. If the proposer does not receive sufficiently many PHASE1B or PHASE2B responses from the acceptors (e.g., because of network partitions or dueling proposers), then the proposer restarts the protocol in a larger round.

Note that it is safe for the leader of round 0 (the smallest round) to skip Phase 1 and proceed directly to Phase 2. Recall that the leader of round $i$ executes Phase 1 to learn of any value that may have already been chosen in any round less than $i$ and to prevent any new values from being chosen in any round less than $i$. There are no rounds less than 0, so these properties are satisfied vacuously.

## 2.3 MultiPaxos

As mentioned earlier, MultiPaxos uses one instance of Paxos for every log entry, choosing the command in the $i$th log entry using the $i$th instance of Paxos. A MultiPaxos deployment that tolerates $f$ faults consists of an arbitrary number of clients, at least $f + 1$ proposers, and $2f + 1$ acceptors (like Paxos), as well as at least $f + 1$ replicas, as illustrated in Figure 3.

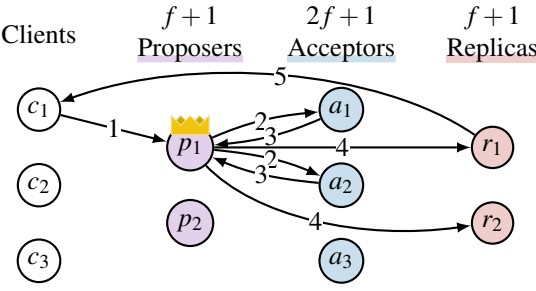

Figure 3: An example execution of MultiPaxos ($f = 1$). The leader is adorned with a crown.

Initially, one of the proposers is elected leader and runs Phase 1 of Paxos for every log entry. Though there are an infinite number of log entries, all but a finite prefix of the log entries are empty, so the leader can run Phase 1 for all entries with only a small number of messages. When a client wants to propose a state machine command $x$, it sends the command to the leader (1). The leader assigns the command a log entry $i$ and then runs Phase 2 of the $i$th Paxos instance to get the value $x$ chosen in entry $i$. That is, the leader sends PHASE2A messages to the acceptors to vote for value $x$ in slot $i$ (2). In the normal case, the acceptors all vote for $x$ in slot $i$ and respond with PHASE2B messages (3). Once the leader learns that a command has been chosen in a given log entry, it informs the replicas (4). Replicas insert commands into their logs and execute the logs in prefix order.

Note that every command is sent to the leader, and the leader performs disproportionately more work per command compared to the other nodes in the protocol. For example, in Figure 3, the leader must send and receive a total of 7 messages per command while the acceptors and replicas send and receive at most 2. This is why the MultiPaxos leader is a well known throughput bottleneck [20, 22].

# 3    Conflict Graphs

## 3.1    Defining Conflict Graphs

By totally ordering state machine commands into a log, state machine replication protocols like MultiPaxos ensure that *every* replica executes *every* command in *exactly the same order*. This is a simple way to ensure that replicas are always in sync, but it is sometimes unnecessary [14]. For example, consider the log shown at the top of Figure 4. The command a=2 (i.e. set the value of variable a to 2) is chosen in log entry 1, and the command b=1 is chosen in log entry 2. With MultiPaxos, every replica would execute these two commands in exactly the same order, but this is not necessary because the commands commute. It is safe for some replicas to execute a=2 before b=1 while other replicas execute b=1 before a=2. The execution order of the two commands has no effect on the final state of the state machine, so they can be safely reordered, as shown in Figure 4.

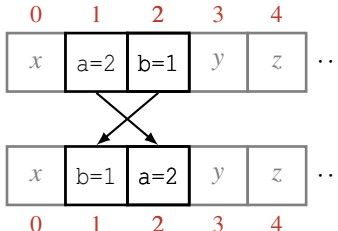

Figure 4: If two commands commute, replicas can safely execute them in either order.

More formally, we say two commands *x* and *y* **conflict** if there exists a state in which executing *x* and then *y* does not produce the same responses or final state as executing *y* and then *x*. We say two commands **commute** if they do not conflict. If two commands conflict (e.g., a=1 and a=2), then they need to be executed by every state machine replica in the same order. But, if two commands commute (e.g., a=2 and b=1), then they do *not* need to be totally ordered. State machine replicas can execute them in either order.

Generalized Multi-leader state machine replication protocols like EPaxos, Caesar, Atlas, and all the BPaxos variants presented in this paper take advantage of command commutativity. Rather than totally ordering commands into a log, these protocols *partially* order commands into a directed graph such that every pair of conflicting commands has an edge between them. We call these graphs **conflict graphs**. An example log and corresponding conflict graph is illustrated in Figure 5. A log consists of a number log entries, and every log entry has a unique log index (e.g., 4). A conflict graph consists of a number of **vertices**, and every vertex has a unique **vertex id** (e.g., $v_4$). Because every vertex is assigned a globally unique vertex id, we often refer to the vertex with vertex id $v$ as $v$. Also note that a command may appear in multiple vertices, in

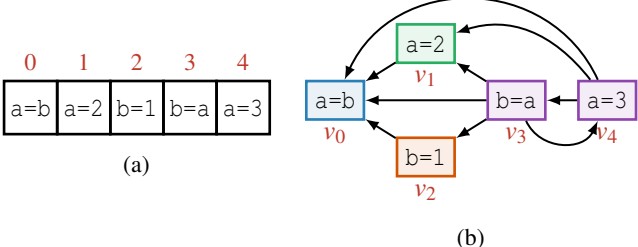

Figure 5: A log and corresponding conflict graph.

much the same way a command may appear multiple times in a log.

Moreover, a vertex $v$ can have directed edges to other vertices. These are called the **dependencies** of $v$, denoted $\text{deps}(v)$. For example, if vertex $v_i$ depends on vertex $v_j$, then there is an edge from $v_i$ to $v_j$. Note that if a pair of commands conflict, then they must have an edge between them. This ensures that every replica executes the two commands in the same order. For example in Figure 5, the commands a=b ($v_0$) and a=2 ($v_1$) conflict, so they have an edge between them. If two commands commute, then they do not have an edge between them. This allows replicas to execute the commands in either oder. For example, the commands a=2 ($v_1$) and b=1 ($v_2$) commute, so there is no edge between them. Finally note that some conflicting commands (e.g., b=a ($v_3$) and a=3 ($v_4$)) have edges in both directions, forming a cycle. Ideally, conflict graphs would be acyclic, but cycles are sometimes unavoidable. The reason for this will become clear soon.

## 3.2    Executing Conflict Graphs

We now explain how to execute a static conflict graph. In the next subsection, we explain how to execute a dynamic conflict graph that grows over time. Replicas execute logs in prefix order. Replicas execute conflict graphs in reverse topological order, one strongly connected component at a time. The order of executing commands within a strongly connected component is not important, but every replica must choose the same order. For example, replicas can execute commands within a component sorted by their vertex id. The conflict graph in Figure 5 has four strongly connected components, each shaded a different color. Vertices $v_0$, $v_1$, and $v_2$ are each in their own components, and commands $v_3$ and $v_4$ are in their own component. Replicas execute these four strongly connected components in reverse topological order as follows:

- First, replicas execute a=b ($v_0$).

- Next, replicas either execute a=2 ($v_1$) then b=1 ($v_2$) or b=1 ($v_2$) then a=2 ($v_1$). There are no edges between vertex $v_1$ and vertex $v_2$, so every replica can execute the two vertices in either order.

Table 1: The differences between protocols like MultiPaxos and Raft that organize commands in logs and protocols like EPaxos, Caesar, and Atlas that organize commands in graphs.

|  | Logs | Graphs |
| --- | --- | --- |
| data structure | log | conflict graph |
|  | log entry | vertex |
|  | log index (e.g., 4) | vertex id (e.g., $v_4$) |
|  | total order | partial order |
| execution order | log order | reverse topological order |
| what's chosen? | commands | commands & dependencies |

- Finally, replicas execute b=a ($v_3$) and a=3 ($v_4$) in some arbitrary but fixed order. For example, if replicas execute commands sorted by their vertex ids, then the replicas would all execute $v_3$ and then $v_4$.

Executing commands in this way, state machine replicas are guaranteed to remain in sync. Every replica executes conflicting commands in the same order, but are free to execute commuting commands in any order.

## 3.3 Constructing Conflict Graphs

In the previous subsection, we explained how to execute a static conflict graph. In reality, graphs are dynamic and grow over time. MultiPaxos constructs one log entry at a time. It uses one instance of consensus for every log entry $i$ to choose which command should be placed in log entry $i$. Analogously, generalized multi-leader protocols construct a conflict graph one vertex at a time. They use one instance of consensus for every vertex $v$ to choose which command should be placed in vertex $v$ and what dependencies, or outbound edges, $v$ should have.

In Figure 6, we illustrate an example execution of how the conflict graph from Figure 5 could be constructed over time. Figure 6 also shows an analogous execution in which a log is constructed over time. Note that a vertex $v$ can be chosen with dependencies deps($v$) before every vertex in deps($v$) has itself been chosen. For example in Figure 6c, $v_3$ is chosen with deps($v_3$) = $\{v_0, v_1, v_2, v_4\}$ before vertices $v_2$ and $v_4$ are chosen. This is analogous to how a command is chosen in log entry 3 in Figure 6h before a command is chosen in entry 2.

A summary of the differences between logs and graphs is given in Table 1.

## 3.4 Two Key Invariants

Protocols like EPaxos, Caesar, Atlas, and the BPaxos protocols in this paper all differ in how they assign commands to vertices, how they compute dependencies, how they implement consensus, and so on. Despite the differences, all the protocols construct conflict graphs one vertex at a time, choosing a command and a set of dependencies $(x, \text{deps}(v))$

for every vertex $v$. The protocols all rely on the following two key invariants for correctness. We call these the **consensus invariant** (Invariant 1) and the **dependency invariant** (Invariant 2).

**Invariant 1** (Consensus Invariant). Consensus is implemented for every vertex $v$. That is, at most one value $(x, \text{deps}(v))$ is chosen for every vertex $v$.

**Invariant 2** (Dependency Invariant). If $(x, \text{deps}(v_x))$ is chosen in vertex $v_x$ and $(y, \text{deps}(v_y))$ is chosen in instance $v_y$, and if $x$ and $y$ conflict, then either $v_x \in \text{deps}(v_y)$ or $v_y \in \text{deps}(v_x)$ or both. That is, if two chosen commands conflict, there is an edge between them.

The consensus invariant ensures that replicas always agree on the state of the conflict graph. It makes it impossible, for example, for two replicas to disagree on which command is in a vertex or disagree on what dependencies a vertex has. The dependency invariant ensures that replicas execute conflicting commands in the same order but does not require that replicas execute commuting commands in the same order. These two invariants are sufficient to ensure linearizable execution. Intuitively, the history of command execution is equivalent to a serial history following any reverse topological ordering of the conflict graph. In fact, replicas literally do execute commands serially according to one of the reverse topological orderings. For a more formal proof, refer to [14] and [21].

## 4 Simple BPaxos

In this section, we introduce **Simple BPaxos**, an inefficient protocol that is designed to be easy to understand. By understanding Simple BPaxos, we will understand of the core mechanisms and invariants that are common to all generalized multi-leader protocols.

### 4.1 Overview

As illustrated in Figure 7, a Simple BPaxos deployment consists of a number of clients, a set of at least $f + 1$ Paxos proposers, a set of $2f + 1$ **dependency service nodes**, a set of $2f + 1$ Paxos acceptors, and a set of at least $f + 1$ replicas. These nodes have the following responsibilities.

- The dependency service nodes, collectively called the **dependency service**, compute dependencies and maintain the dependency invariant (Invariant 2).

- The proposers and acceptors implement one instance of Paxos for every vertex and maintain the consensus invariant (Invariant 1).

- The replicas construct and execute conflict graphs and send the results of executing commands back to the clients.

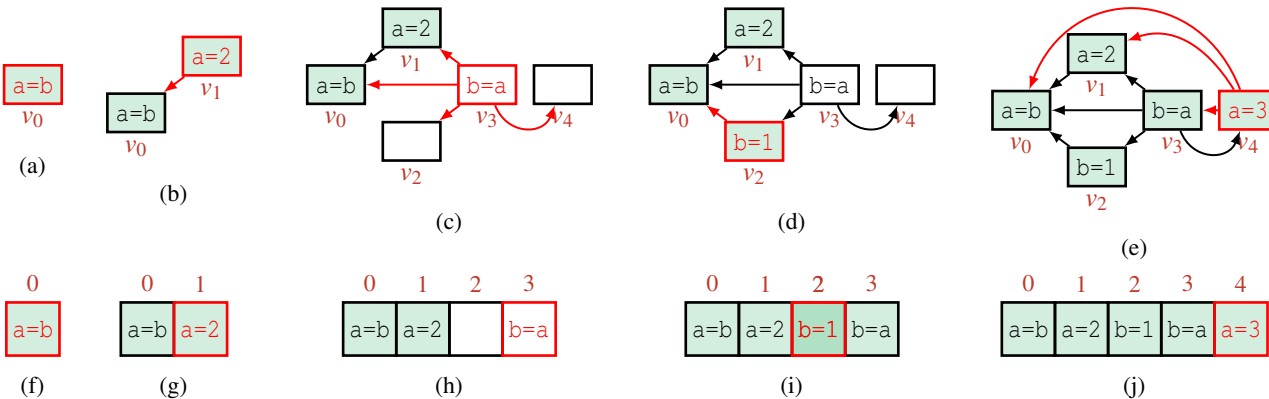

Figure 6: In subfigures (a) - (e), we see a conflict graph constructed over time. The most recently chosen vertex is drawn in red. The executed commands are shaded green. (a) The command a=b is chosen in vertex $v_0$ without any dependencies. The command is executed immediately. (b) The command a=2 is chosen in vertex $v_1$ with a dependency on $v_0$. The command is executed immediately. (c) The command b=a is chosen in vertex $v_3$ with dependencies on $v_0$, $v_1$, $v_2$, and $v_4$. No commands have been chosen in $v_2$ and $v_4$ yet, so $v_3$ cannot be executed. (d) The command b=1 is chosen in vertex $v_2$ with a dependency on $v_0$. The command is executed immediately. (e) The command a=3 is chosen in vertex $v_4$ with dependencies on $v_0$, $v_1$, and $v_3$. Now $v_3$ and $v_4$ are executed. In subfigures (f) - (j), we see an analogous execution for a log.

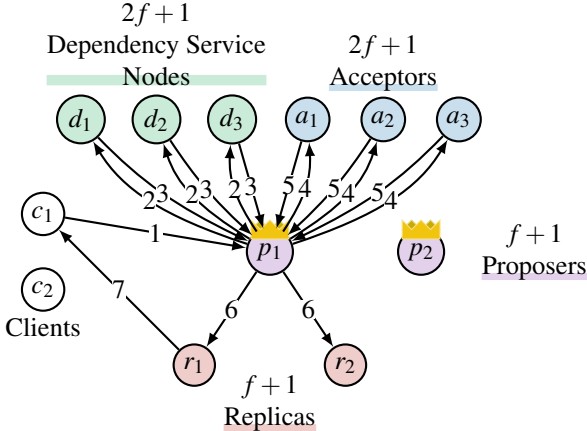

Figure 7: An example execution of Simple BPaxos ($f = 1$).

More concretely, Simple BPaxos executes as follows. The numbers here correspond to the numbered arrows in Figure 7.

- **(1)** When a client wants to propose a state machine command $x$, it sends $x$ to *any* of the proposers. Note that with MultiPaxos, only one proposer is elected leader, but in Simple BPaxos, every proposer is a leader.

- **(2) and (3)** When a proposer $p_i$ receives a command $x$, from a client, it places $x$ in a vertex with globally unique vertex id $v_x = (p_i, m)$ where $m$ is a monotonically increasing integer local to $p_i$. For example, proposer $p_i$ places the first command that it receives in vertex $(p_i, 0)$, the next command in vertex $(p_i, 1)$, the next in $(p_i, 2)$, and so on. The proposer then performs a round trip of

communication with the dependency service. It sends $v_x$ and $x$ to the dependency service, and the dependency service replies with the dependencies $\text{deps}(v_x)$. For now, we leave this process abstract. We'll explain how the dependency service computes dependencies in Section 4.2.

- **(4) and (5)** The proposer $p_i$ then executes Phase 2 of Paxos with the acceptors, proposing that the value $(x, \text{deps}(v_x))$ be chosen in the instance of Paxos associated with vertex $v_x = (p_i, m)$. This is analogous to a MultiPaxos leader running Phase 2, proposing the command $x$ be chosen in the instance of Paxos associated with log entry $m$.

  Recall from Section 2 that the Paxos proposer executing round 0 can safely bypass Phase 1. By design, we predetermine that the proposer $p_i$ leads round 0 for vertices of the form $(p_i, m)$. This is why $p_i$ can safely bypass Phase 1 and immediately execute Phase 2.

  In the normal case, $p_i$ gets the value $(x, \text{deps}(v_x))$ chosen in vertex $v_x$. It is also possible that some other proposer erroneously concluded that $p_i$ had failed and proposed some other value in vertex $v_x$, but we discuss this scenario later.

- **(6)** The proposer $p_i$ broadcasts $v_x$, $x$, and $\text{deps}(v_x)$ to all of the replicas. The replicas add vertex $v_x$ to their conflict graph with command $x$ and with edges to the vertices in $\text{deps}(v_x)$. The replicas execute their conflict graphs as described in Section 3.

- **(7)** Once a replica executes command $x$, it sends the result of executing command $x$ back to the client.

## 4.2 Dependency Service

The dependency service consists of $2f + 1$ dependency service nodes $d_1, \ldots, d_{2f+1}$. Every dependency service node maintains an acyclic conflict graph. These conflict graphs are similar but not equal to the conflict graph that Simple BPaxos ultimately executes.

When a proposer sends a vertex $v_x$ with command $x$ to the dependency service, it sends $v_x$ and $x$ to every dependency service node. When a dependency service node $d_i$ receives $v_x$ and $x$, it performs one of the following two actions depending on whether $d_i$'s graph already contains vertex $v_x$.

- If $d_i$'s conflict graph does not contain vertex $v_x$, then $d_i$ adds vertex $v_x$ to its graph with command $x$. $d_i$ adds an edge from $v_x$ to every other vertex $v_y$ with command $y$ if $x$ and $y$ conflict. Letting $\text{out}(v_x)$ be the set of vertices to which $v_x$ has an edge, $d_i$ then returns $\text{out}(v_x)$ to the proposer.

- Otherwise, if $d_i$'s conflict graph already contains vertex $v_x$, then $d_i$ does not modify its conflict graph. It immediately returns $\text{out}(v_x)$ to the proposer.

An example execution of a dependency service node is given in Figure 8.

When a proposer receives replies from $f + 1$ dependency service nodes, it takes the union of these responses as the value of $\text{deps}(v_x)$. For example, imagine $f = 1$ and a proposer receives dependencies $\{v_w, v_y\}$ from $d_1$ and dependencies $\{v_w, v_z\}$ from $d_2$. The proposer computes $\text{deps}(v_x) = \{v_w, v_y, v_z\}$. The dependency service maintains Invariant 3.

**Invariant 3.** If two conflicting commands $x$ and $y$ in vertices $v_x$ and $v_y$ yield dependencies $\text{deps}(v_x)$ and $\text{deps}(v_y)$ from the dependency service, then either $v_x \in \text{deps}(v_y)$ or $v_y \in \text{deps}(v_x)$ or both.

*Proof.* Consider conflicting commands $x$ and $y$ in vertices $v_x$ and $v_y$ with dependencies $\text{deps}(v_x)$ and $\text{deps}(v_y)$ computed by the dependency service. $\text{deps}(v_x)$ is the union of dependencies computed by $f + 1$ dependency service nodes $D_x$. Similarly, $\text{deps}(v_y)$ is the union of dependencies computed by $f + 1$ dependency service nodes $D_y$. Because $f + 1$ is a majority of $2f + 1$, $D_x$ and $D_y$ necessarily intersect. That is, there is some dependency service node $d_i$ that is in $D_x$ and $D_y$. $d_i$ either received $v_x$ or $v_y$ first. If it received $v_x$ first, then it returns $v_x$ as a dependency of $v_y$, so $v_x \in \text{deps}(v_y)$. If it received $v_y$ first, then it returns $v_y$ as a dependency of $v_x$, so $v_y \in \text{deps}(v_x)$. $\square$

## 4.3 An Example

An example execution of Simple BPaxos with $f = 1$ is illustrated in Figure 9.

- In Figure 9a, proposer $p_1$ receives command $x$ from a client, while proposer $p_2$ receives command $y$ from a

client. The commands are placed in vertices $v_x$ and $v_y$ respectively.

- In Figure 9b, $p_1$ sends $x$ in $v_x$ to the dependency service, while $p_2$ concurrently sends $y$ in $v_y$. Dependency service nodes $d_1$ and $d_2$ receive $x$ and then $y$, so they compute $\text{deps}(v_x) = \emptyset$ and $\text{deps}(v_y) = \{v_x\}$. $d_3$, on the other hand, receives $y$ and then $x$ and computes $\text{deps}(v_x) = \{v_y\}$ and $\text{deps}(v_y) = \emptyset$

  $p_1$ receives $\emptyset$ from $d_2$ and $\{v_y\}$ from $d_3$. Two dependency service nodes form a majority, so $p_1$ computes $\text{deps}(v_x) = \{v_y\} \cup \emptyset = \{v_y\}$. Similarly, $p_2$ receives $\{v_x\}$ from $d_2$ and $\emptyset$ from $d_3$, so $p_2$ computes $\text{deps}(v_y) = \{v_x\} \cup \emptyset = \{v_x\}$. Note that $p_1$ and $p_2$ also receive responses from $d_1$, but proposers form dependencies from the first set of $f + 1$ dependency service nodes they hear from.

- In Figure 9c, $p_1$ executes Phase 2 of Paxos to get the value $(x, \{v_y\})$ chosen in vertex $v_x$. $p_2$ likewise gets the value $(y, \{v_x\})$ chosen in vertex $v_y$.

- In Figure 9d, the proposers broadcast their commands to the replicas. The replicas add $v_x$ and $v_y$ to their conflict graphs and execute the commands once they have received both. One or more of the replicas also sends the results of executing $x$ and $y$ back to the clients.

Note that the replicas' conflict graphs contain a cycle. This is because the dependency service nodes do not receive every command in the same order. In Figure 9, dependency service nodes $d_2$ and $d_3$ receive $x$ and $y$ in opposite orders, leading to the two commands depending on each other. It is tempting to enforce that every dependency service node receive every command in exactly the same order, but unfortunately, this would be tantamount to solving consensus [6].

## 4.4 Recovery

Imagine a proposer receives a command $x$ from a client, places the command $x$ in vertex $v_x$, sends $v_x$ and $x$ to the dependency service, and then crashes. Because a command and a set of dependencies have not been chosen in vertex $v_x$ yet, we call $v_x$ unchosen. It is possible that a command $y$ chosen in vertex $v_y$ depends on an unchosen vertex $v_x$. If vertex $v_x$ remains forever unchosen, then the command $y$ will never be executed. To avoid this liveness violation, if any replica notices that vertex $v_x$ has been unchosen for some time, it notifies a proposer. The proposer then executes Phase 1 and Phase 2 of Paxos with the acceptors to get a **noop** chosen in vertex $v_x$ without any dependencies. noop is a distinguished command that does not affect the state machine and does not conflict with any other command. An example of this execution is given in Figure 10.

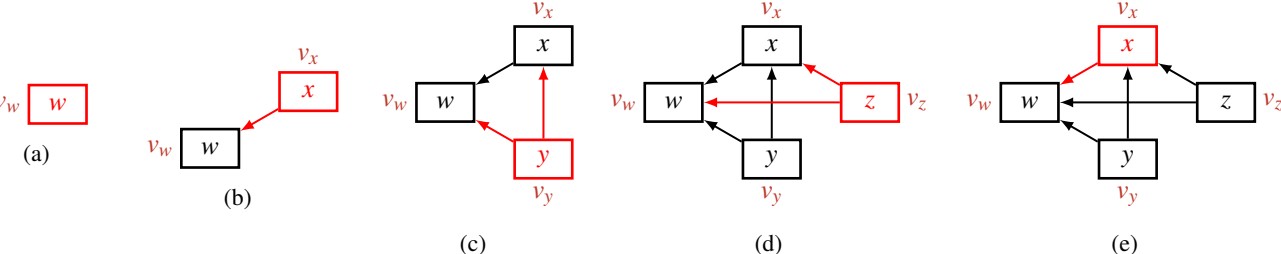

Figure 8: In subfigures (a) – (e), we see the execution of a dependency service node $d_i$. (a) $d_i$ receives command $w$ in vertex $v_w$. $d_i$ adds this vertex to its conflict graph and because there are no other vertices, it returns the dependencies $\text{deps}(v_w) = \emptyset$. (b) $d_i$ receives command $x$ in vertex $v_x$. $d_i$ adds this vertex to its conflict graph. $x$ conflicts with $w$, so $d_i$ adds an edge from $v_x$ to $v_w$ and returns the dependencies $\text{deps}(v_x) = \{v_w\}$. (c) $d_i$ receives command $y$ in vertex $v_y$. $d_i$ adds this vertex to its conflict graph. $y$ conflicts with $w$ and $x$, so $d_i$ adds an edge from $v_y$ to $v_w$ and from $v_y$ to $v_x$. It returns the dependencies $\text{deps}(v_y) = \{v_w, v_x\}$. (d) $d_i$ receives command $z$ in vertex $v_z$. $d_i$ adds this vertex to its conflict graph. $z$ conflicts with $w$ and $x$, so $d_i$ adds an edge from $v_z$ to $v_w$ and from $v_z$ to $v_x$. It returns the dependencies $\text{deps}(v_z) = \{v_w, v_x\}$. (e) $d_i$ receives command $x$ in vertex $v_x$. $d_i$'s graph already contains vertex $v_x$, so $d_i$ returns the dependencies $\text{deps}(v_x) = \{v_w\}$ and does not modify its graph.

- In Figure 10a, proposer $p_1$ receives command $x$ from a client. It places $x$ in vertex $v_x$ and sends $v_x$ and $x$ to the dependency service. Shortly after, it fails.

- In Figure 10b, proposer $p_2$ receives command $y$ from a client. It places $y$ in $v_y$ and contacts the dependency service. The dependency service nodes have already received $x$ in $v_x$, so they compute $\text{deps}(v_y) = \{v_x\}$. $p_2$ then gets $y$ chosen in vertex $v_y$ with a dependency on $v_x$ and broadcasts it to the replicas.

- In Figure 10c, the replicas cannot execute vertex $v_y$ because it depends on the unchosen vertex $v_x$. After a timeout expires, replica $r_1$ notifies $p_2$ that the vertex has been unchosen for some time.

- In Figure 10d, $p_2$ executes Phase 1 and Phase 2 of Paxos in some round $r > 0$ with the acceptors to get the command noop chosen in vertex $v_x$ without any dependencies. $p_2$ notifies the replicas, and the replicas place the noop in vertex $v_x$. The replicas execute their conflict graphs in reverse topological order. They execute the noop first (which has no effect) and then execute $y$.

  Note that $p_2$ must execute both phases of Paxos because it is not in round 0. This is necessary to ensure that no other value could have been chosen in $v_x$.

Note that a Simple BPaxos proposer recovers a command and proposes a noop by executing Paxos as normal. Simple BPaxos does not require an additional recovery protocol. Rather, commands and noops are proposed in the exact same way. This simplifies the protocol.

Also note that if a client does not receive a response for its pending request for a sufficiently long period of time, it resends its request. This means that if a client's command is replaced by a noop, the client will eventually re-propose the command.

## 4.5 Safety

To ensure that Simple BPaxos is safe, we must ensure that it maintains the consensus invariant and the dependency invariant. Simple BPaxos maintains the consensus invariant because it implements Paxos. The dependency invariant follows immediately from Invariant 3 and the fact that noops don't conflict with any other command.

## 5 Fast Paxos

Simple BPaxos is designed to be easy to understand, but as shown in Figure 9, it takes seven network delays (in the best case) between when a client proposes a command $x$ and when a client receives the result of executing $x$. Call this duration of time the **commit time**. Generalized multi-leader protocols like EPaxos, Caesar, and Atlas all achieve a commit time of only four network delays in the best case. They do so by leveraging Fast Paxos [15].

Fast Paxos is a Paxos variant that allows clients to propose values directly to the acceptors without having to initially contact a proposer. Fast Paxos is an optimistic protocol. If all of the acceptors happen to receive the same command from the clients, then Fast Paxos has a commit time of only three network delays. This is called the fast path. However, if multiple clients concurrently propose different commands, and not all of the acceptors receive the same command, then the protocol reverts to a slow path and introduces two additional network delays to the commit time. In this section, we review a slightly simplified version of Fast Paxos.

### 5.1 Overview

Like Paxos, a Fast Paxos deployment consists of some number of clients, $f + 1$ proposers, and $2f + 1$ acceptors. We also

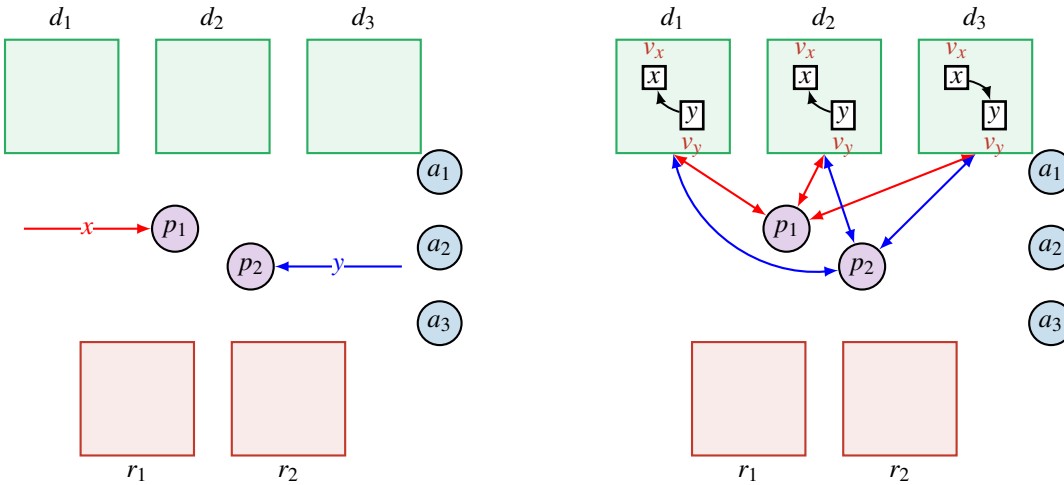

(a) $p_1$ receives command $x$; $p_2$ receives command $y$.

(b) The proposers contact the dependency service.

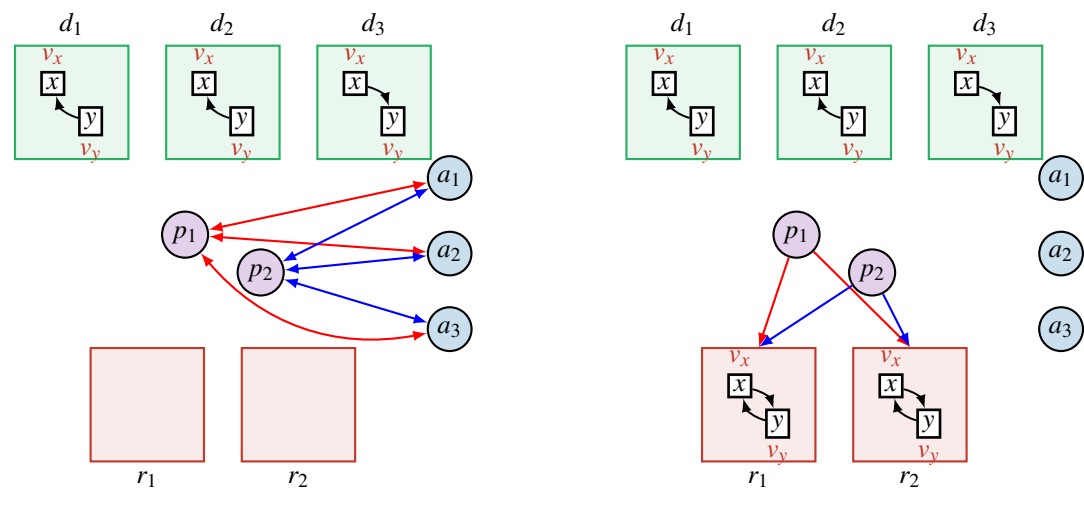

(c) The proposers contact the acceptors.

(d) The proposers notify the replicas.

Figure 9: An example execution of Simple BPaxos ($f = 1$).

include a set of $f + 1$ **learners**. These nodes are notified of the value chosen by Fast Paxos. Note that we use the term learner rather than replica because Fast Paxos is a consensus protocol and not a state machine replication protocol, so there are no state machine replicas. A Fast Paxos deployment is illustrated in Figure 11. Proposer and acceptor pseudocode are given in Algorithm 1 and Algorithm 2.

Like Paxos, Fast Paxos is divided into a number of integer valued rounds. The key difference is that round 0 of Fast Paxos is a special "fast round." A client can propose a value directly to an acceptor in round 0 without having to contact a proposer first. The normal case execution of Fast Paxos is illustrated in Figure 11a. The execution proceeds as follows:

- **(1)** When a client wants to propose a value $v$, it sends $v$ to all of the acceptors.

- **(2)** When an acceptor receives a value $v$ from a client, the acceptor ignores $v$ if it has already received a message in some round $i \geq 0$. Otherwise, it votes for $v$ by updating its state and sending a PHASE2B$\langle 0, v \rangle$ message to the proposer that leads round 0. This is shown in Algorithm 2 line 1 – line 4.

- **(3)** Let maj$(n) = \lceil \frac{n+1}{2} \rceil$ be a majority of $n$. If the proposer that leads round 0 receives PHASE2B$\langle 0, v' \rangle$ messages from $f + $ maj$(f + 1)$ acceptors for the same value $v'$, then $v'$ is chosen, and the proposer notifies the learners. This is shown in Algorithm 1 line 1 – line 3. We consider what happens when not every value is the same in Section 5.2.

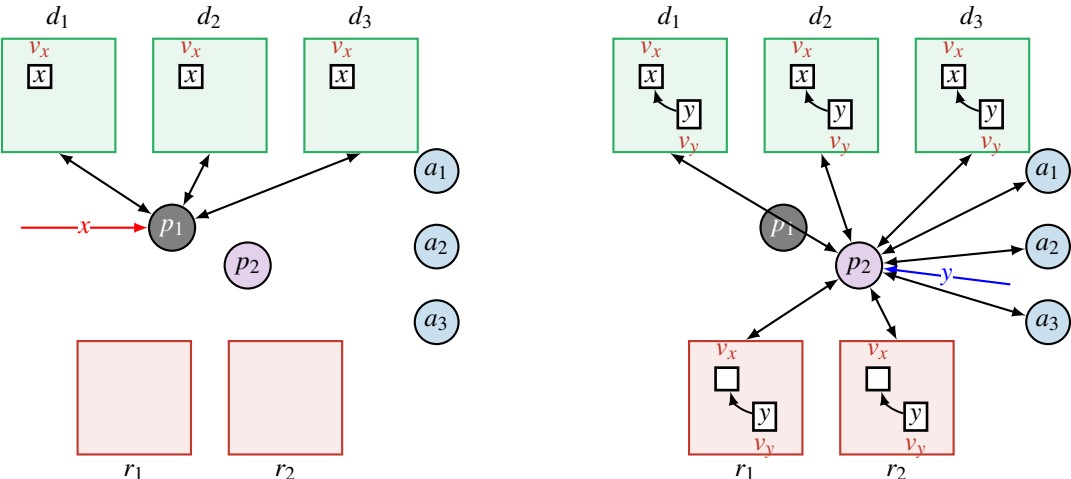

(a) $p_1$ receives $x$, talks to the dependency service, and fails. (b) $p_2$ receives $y$, gets it chosen with a dependency on $v_x$.

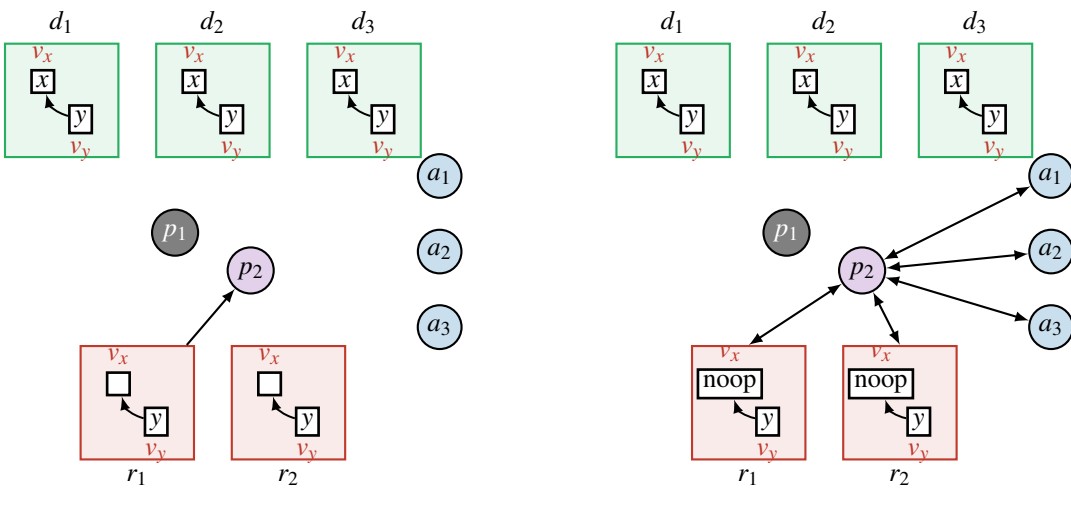

(c) A replica notifies $p_2$ that $v_x$ is unchosen.

(d) $p_2$ gets a noop chosen in $v_x$.

Figure 10: An example execution of Simple BPaxos recovery ($f = 1$).

## 5.2  Recovery

Note that in Paxos, a value is chosen when $f + 1$ acceptors vote for it in some round $i$. In round 0 of Fast Paxos, a value is chosen when $f + \mathsf{maj}(f + 1)$ acceptors vote for it. The larger number of required votes is needed to ensure the safety of recovery, which we now describe.

Let $p$ be the proposer leading round 0. Recovery is the process by which a proposer other than $p$ gets a value chosen. For example, if $p$ fails, some other proposer must take over and get a value chosen. Recovery is illustrated in Figure 11b.

- **(1) and (2)** A recovering proposer performs Phase 1 of Paxos with at least $f + 1$ acceptors in some round $i > 0$. This is shown in Algorithm 1 line 7 – line 9 and Algorithm 2 line 5 – line 7.

- **(3) and (4)** The recovering proposer receives PHASE1B$\langle i, vr, vv \rangle$ messages from $f + 1$ acceptors. Call this quorum of acceptors $A$. The proposer computes $k$ as the largest received $vr$ (line 11). This is the largest round in which any acceptor in $A$ has voted. If $k = -1$ (line 12), then none of the acceptors have voted in any round less than $i$, so the proposer is free to propose an arbitrary value. This is the same as in Paxos. If $k > 0$ (line 14), then the proposer must propose the value $vv$ proposed in round $k$. Again, this is the same as in Paxos. $vv$ may have been chosen in round $k$, so the proposer is forced to propose it as well. If $k = 0$ (line 16), then there are two cases to consider.

First, if $\mathsf{maj}(f + 1)$ of the acceptors in $A$ have all voted for some value $v'$ in round 0, then it's possible that $v'$

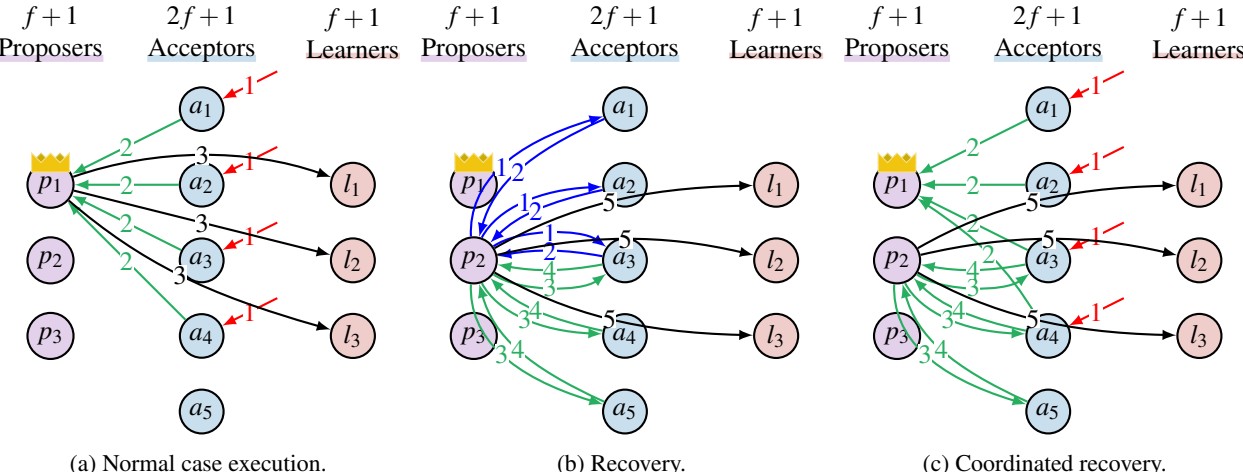

Figure 11: Example executions of Fast Paxos ($f = 2$). The leader of round 0 is adorned with a crown. Client requests are drawn in red. Phase 1 messages are drawn in blue. Phase 2 messages are drawn in green.

was chosen in round 0 (line 17). Specifically, if all $f$ of the acceptors not in $A$ voted for $v'$ in round 0, then along with the $\mathsf{maj}(f+1)$ of acceptors in $A$ who also voted for $v'$ in round 0, there is a quorum of $f + \mathsf{maj}(f+1)$ acceptors who voted for $v'$ in round 0. In this case, the proposer must propose $v'$ as well since it might have been chosen. Second, if there does not exist $\mathsf{maj}(f+1)$ votes for any value $v'$, then the proposer concludes that no value was chosen or every will be chosen in round 0, so it is free to propose an arbitrary value (line 19). Once the recovering proposer determines which value to propose, it gets the value chosen with the acceptors using the normal Phase 2 of Paxos.

Note that a value must receive at least $f + \mathsf{maj}(f+1)$ votes in round 0 to be chosen. If this number were any smaller, it would be possible for a recovering proposer to find two distinct values $v'$ and $v''$ that *both* could have been chosen in round 0. In this case, the proposer cannot make progress. It cannot propose $v'$ because $v''$ might have been chosen, and it cannot propose $v''$ because $v'$ might have been chosen

More concretely, imagine an Fast Paxos deployment with $f = 2$ and five acceptors $a_1$, $a_2$, ..., $a_5$. Further imagine that a value is considered chosen after receiving votes from only 3 (i.e. $f+1$) acceptors rather than the correct number of 4 (i.e. $f + \mathsf{maj}(f+1)$). Consider a proposer executing Phase 1 in round 1. It contacts $a_3$, $a_4$, and $a_5$. $a_3$ voted for value $x$ in round 0; $a_4$ voted for value $y$ in round 0; and $a_5$ didn't vote in round 0. What value should the proposer propose in Phase 2? Well, $x$ was maybe chosen in round 0 (if $a_1$ and $a_2$ both voted for $x$ in round 0), so the proposer has to propose $x$. However, $y$ was also maybe chosen in round 0 (if $a_1$ and $a_2$ both voted for $y$ in round 0), so the proposer

also has to propose $y$. The proposer can only propose one value, so the protocol gets stuck. By requiring $f + \mathsf{maj}(f+1)$ votes rather than $f+1$ votes, we eliminate these situations. It's not possible for two values to both potentially have received $f + \mathsf{maj}(f+1)$ votes. There isn't enough acceptors for this to be possible.

- **(5)** The proposer notifies the learners of the chosen value.

## 5.3  Coordinated Recovery

Finally, we consider what happens when the proposer of round 0 receives $f + \mathsf{maj}(f+1)$ PHASE1B messages from the acceptors, but without all of them containing the same value $v'$. Naively, the proposer could simply perform a recovery, executing both phases of Paxos is some round $r > 0$. However, if we assign rounds to proposers in such a way that the proposer of round 0 is also the proposer of round 1, then we can take advantage of an optimization called **coordinated recovery**. This is illustrated in Figure 11c and proceeds as follows:

- **(1)** Multiple clients send distinct commands directly to the acceptors.

- **(2)** The acceptors receive and vote for the commands and send PHASE2B messages to the leader of round 0. However, not every acceptor receives the same value first, so not all the acceptors vote for the same value.

- **(3) and (4)** The proposer receives PHASE2B messages from $f + \mathsf{maj}(f+1)$ acceptors, but the acceptors have not all voted for the same value. At this point, the proposer could naively perform a recovery in round 1 by executing Phase 1 and then Phase 2 of Paxos. But, executing Phase 1 in round 1 is redundant, since the PHASE2B

**Algorithm 1** Fast Paxos Proposer

**State:** a value $v$, initially null
**State:** a round $i$, initially $-1$
 1: **upon** receiving PHASE2B$\langle 0, v' \rangle$ from $f + \mathsf{maj}(f + 1)$ acceptors as the proposer of round 0 with $i = 0$ **do**
 2:     **if** every value of $v'$ is the same **then**
 3:         $v'$ is chosen, notify the learners
 4:     **else**
 5:         $i \leftarrow 1$
 6:         proceed to line 11 viewing every PHASE2B$\langle 0, v' \rangle$ as a PHASE1B$\langle 1, 0, v' \rangle$
 7: **upon** recovery **do**
 8:     $i \leftarrow$ next largest round owned by this proposer
 9:     send PHASE1A$\langle i \rangle$ to the acceptors
10: **upon** receiving PHASE1B$\langle i, vr, vv \rangle$ from $f + 1$ acceptors **do**
11:     $k \leftarrow$ the largest $vr$ in any PHASE1B$\langle i, vr, vv \rangle$
12:     **if** $k = -1$ **then**
13:         $v \leftarrow$ an arbitrary value
14:     **else if** $k > 0$ **then**
15:         $v \leftarrow$ the corresponding $vv$ in round $k$
16:     **else if** $k = 0$ **then**
17:         **if** there are $\mathsf{maj}(f + 1)$ PHASE1B$\langle i, 0, v' \rangle$ messages for some value $v'$ **then**
18:             $v \leftarrow v'$
19:         **else**
20:             $v \leftarrow$ an arbitrary value
21:     send PHASE2A$\langle i, v \rangle$ to the acceptors
22: **upon** receiving PHASE2B$\langle i \rangle$ from $f + 1$ acceptors **do**
23:     $v$ is chosen, notify the learners

---

messages in round 0 contain exactly the same information as the PHASE1B messages in round 1. Specifically, the proposer can view every PHASE2B$\langle 0, v' \rangle$ message as a proxy for a PHASE1B$\langle 1, 0, v' \rangle$ message. Thus, the proposer instead jumps immediately to Phase 2 in round 1 to get a value chosen (line 4 – line 6).

- **(5)** Finally, the proposer notifies the learners of the chosen value.

# 6 Fast BPaxos

In this section, we present a purely pedagogical protocol called **Fast BPaxos**. Fast BPaxos achieves a commit time of four network delays (compared to Simple BPaxos' seven), but Fast BPaxos is unsafe. It does not properly implement state machine replication. To understand why more complex protocols like EPaxos, Caesar, and Atlas work the way they do, it helps to understand why simpler protocols like Fast BPaxos don't work in the first place. Understanding why Fast BPaxos is unsafe leads to fundamental insights into these

**Algorithm 2** Fast Paxos Acceptor

**State:** the largest seen round $r$, initially $-1$
**State:** the largest round $vr$ voted in, initially $-1$
**State:** the value $vv$ voted for in round $vr$, initially null
 1: **upon** receiving value $v$ from client **do**
 2:     **if** $r = -1$ **then**
 3:         $r, vr, vv \leftarrow 0, 0, v$
 4:         send PHASE2B$\langle 0, v \rangle$ to proposer of round 0
 5: **upon** receiving PHASE1A$\langle i \rangle$ from $p$ with $i > r$ **do**
 6:     $r \leftarrow i$
 7:     send PHASE1B$\langle i, vr, vv \rangle$ to $p$
 8: **upon** receiving PHASE2A$\langle i, v \rangle$ from $p$ with $i \geq r$ **do**
 9:     $r, vr, vv \leftarrow i, i, v$
10:     send PHASE2B$\langle i \rangle$ to $p$

---

other protocols.

## 6.1 The Protocol

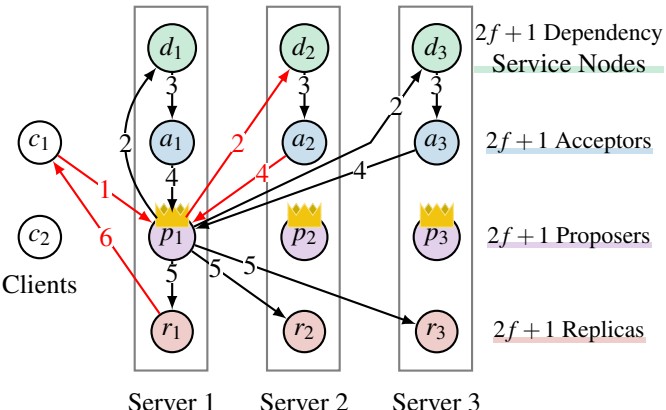

Figure 12: An example execution of Fast BPaxos ($f = 1$). The four network delays are drawn in red.

Fast BPaxos is largely identical to Simple BPaxos with one key observation. Rather than implementing Paxos, Fast BPaxos implements Fast Paxos. This allows the protocol to reduce the commit time by overlapping communication with the dependency service (to compute dependencies) and communication with the acceptors (to implement consensus).

As shown in Figure 12, a Fast BPaxos deployment consists of $2f + 1$ dependency service nodes, $2f + 1$ Fast Paxos acceptors, $2f + 1$ Fast Paxos proposers, and $2f + 1$ replicas. These logical nodes are co-located on a set of $2f + 1$ servers, where every physical server executes one dependency service node, one acceptor, one proposer, and one replica. For example, in Figure 12, server 2 executes $d_2$, $a_2$, $p_2$, and $r_2$. As illustrated in Figure 12, the protocol executes as follows:

- **(1)** When a client wants to propose a state machine com-

mand $x$, it sends $x$ to *any* of the proposers.

- **(2)** When a proposer $p_i$ receives a command $x$, from a client, it places $x$ in a vertex with globally unique vertex id $v_x = (p_i, m)$ where $m$ is a monotonically increasing integer local to $p_i$. $p_i$ then sends $v_x$ and $x$ to all of the dependency service nodes. Note that we predetermine that proposer $p_i$ is the leader of round 0 and 1 of the Fast Paxos instance associated with vertex $v_x = (p_i, m)$.

- **(3)** When a dependency service node $d_j$ receives a command $x$ in vertex $v_x$, it computes a set of dependencies $\text{deps}(v_x)$ in the exact same way as in Simple BPaxos (i.e. $d_j$ maintains an acyclic conflict graph). Unlike Simple BPaxos however, $d_j$ does not send $\text{deps}(v_x)$ back to the proposer. Instead, it proposes to the co-located Fast Paxos acceptor $a_j$ that the value $(x, \text{deps}(v_x))$ be chosen in the instance of Fast Paxos associated with vertex $v_x$ in round 0.

- **(4)** Fast BPaxos acceptors are unmodified Fast Paxos acceptors. In the normal case, when an acceptor $a_j$ receives value $(x, \text{deps}(v_x))$ in vertex $v_x = (p_i, m)$, it votes for the value and sends the vote to $p_i$.

- **(5)** Fast BPaxos proposers are unmodified Fast Paxos proposers. In the normal case, $p_i$ receives $f + \text{maj}(f + 1)$ votes for value $(x, \text{deps}(v_x))$ in vertex $v_x$, so $(x, \text{deps}(v_x))$ is chosen. The proposer broadcasts the command and dependencies to the replicas. If $p_i$ receives $f + \text{maj}(f + 1)$ votes, but they are not all for the same value, the proposer executes coordinate recovery (see Algorithm 1 line 4 – line 6).

- **(6)** Fast BPaxos replicas are identical to Simple BPaxos replicas. Replicas maintain and execute conflict graphs, returning the results of executing commands to the clients.

Note that Figure 12 illustrates six steps of execution, but the commit time is only four network delays (drawn in red). Communication between co-located components (e.g., between $d_1$ and $a_1$ and between $p_1$ and $r_1$) does not involve the network and therefore does not contribute to the commit time.

## 6.2   Recovery

As with Simple BPaxos, it is possible that a command $y$ chosen in vertex $v_y$ depends on an unchosen vertex $v_x$ that must be recovered for execution to proceed. Fast BPaxos performs recovery in the same way as Simple BPaxos. If a replica detects that a vertex $v_x$ has been unchosen for some time, it notifies a proposer. The proposer then executes a Fast Paxos recovery to get a noop chosen in vertex $v_x$ with no dependencies.

## 6.3   Lack of Safety

We now demonstrate why Fast BPaxos is unsafe. Consider the execution of Fast BPaxos ($f = 2$) illustrated in Figure 13.

- In Figure 13a, proposer $p_1$ receives command $x$ from a client. It places $x$ in vertex $v_x$ and sends $v_x$ and $x$ to the dependency service. $d_1$ and $d_2$ receive the message. They place $x$ in their conflict graphs without any dependencies, and send the value $(x, \emptyset)$ to their co-located acceptors. $a_1$ and $a_2$ vote for $(x, \emptyset)$ in vertex $v_x$, but $p_1$ crashes before it receives any of these votes. The messages sent to $d_3$, $d_4$, and $d_5$ are dropped by the network.

- Similarly in Figure 13a, proposer $p_5$ receives a conflicting command $y$, $p_5$ sends $v_y$ and $y$ to $d_4$ and $d_5$, $d_4$ and $d_5$ propose $(y, \emptyset)$ to $a_4$ and $a_5$, $a_4$ and $a_5$ vote for the proposed values, and $p_5$ fails.

- In Figure 13b, $p_2$ recovers vertex $v_x$. To recover $v_x$, $p_2$ executes Phase 1 of Fast Paxos with acceptors $a_1$, $a_2$, and $a_3$. $p_2$ observes that $a_1$ and $a_2$ both voted for the value $(x, \emptyset)$ in round 0. Therefore, $p_2$ concludes that $(x, \emptyset)$ may have been chosen in round 0, so it proceeds to Phase 2 and gets the value $(x, \emptyset)$ chosen in vertex $v_x$ (Algorithm 1 line 17). $p_2$ cannot propose any other value (e.g., a noop) because $(x, \emptyset)$ may have already been chosen. This is a core invariant of Paxos. From our omniscient view of the execution, we know that $(x, \emptyset)$ was never chosen, but from $p_2$'s myopic view, it cannot make this determination and so is forced to propose $(x, \emptyset)$. This is a **critical point** in the execution, which we will discuss further in a moment.

- In Figure 13b, proposer $p_4$ recovers vertex $v_y$ in much the same way as $p_2$ recovers $v_x$. $p_4$ observes that $a_4$ and $a_5$ voted for $(y, \emptyset)$ in round 0, so it is forced to get the value $(y, \emptyset)$ chosen.

- Finally in Figure 13b, proposers $p_2$ and $p_4$ broadcast $(x, \emptyset)$ and $(y, \emptyset)$ to all of the replicas. The replicas place $x$ and $y$ in their conflict graphs without edges between them. This violates the dependency invariant. $x$ and $y$ conflict, so there must be an edge between them. Without an edge, the replicas can execute $x$ and $y$ in different orders, causing their states to diverge.

What went wrong? When $p_2$ was recovering $v_x$, Fast Paxos forced it to choose $(x, \emptyset)$. However, the dependencies $\text{deps}(v_x) = \emptyset$ were *not* computed by a majority of the dependency service nodes. They were computed only by $d_1$ and $d_2$. This is what allowed conflicting commands $x$ and $y$ to be chosen without a dependency on each other.

This example illustrates a **fundamental tension** between preserving the consensus invariant (Invariant 1) and preserving the dependency invariant (Invariant 2). Maintaining the consensus invariant in isolation is easy (e.g., use Paxos), and

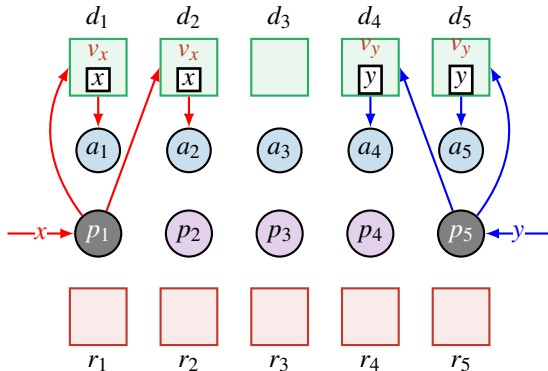

(a) $p_1$ receives $x$, talks to the dependency service, and fails. $p_2$ receives $y$, talks to the dependency service, and fails.

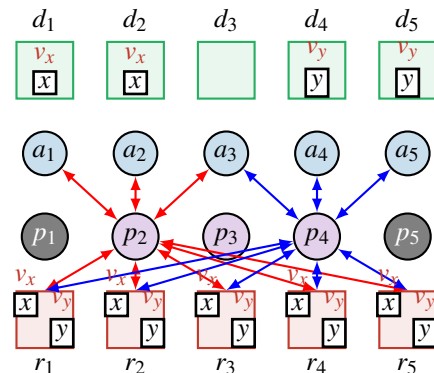

(b) $p_2$ recovers $v_x$ with command $x$ and no dependencies. $p_4$ recovers $v_y$ with command $y$ and no dependencies.

Figure 13: A Fast BPaxos bug ($f = 2$). Conflicting commands $x$ and $y$ are executed in different orders by different replicas.

maintaining the dependency invariant in isolation is also easy (e.g., use the dependency service). But, maintaining both invariants simultaneously is tricky.

When performing a recovery, like the one in our example above, a proposer is sometimes forced to propose a particular value (e.g., $(x, \emptyset)$) in order to properly implement consensus and simultaneously forced *not* to propose the value in order to correctly compute dependencies. Resolving the tension between the consensus and dependency invariants during recovery is the single most important and the single most challenging aspect of generalized multi-leader protocols like EPaxos, Caesar, and Atlas. All of these protocols have a similar structure and execution on the normal path. They all compute dependencies from a majority of servers, and they all execute Fast Paxos variants to get these dependencies chosen. If you understand the normal case execution of one of these protocols, it is not difficult to understand the others. The key feature that distinguishes the protocols is how they resolve the fundamental tension between implementing consensus and computing dependencies.

These protocols all take different approaches to resolving the tension. In the next two sections, we broadly categorize the approaches into two main techniques: *tension avoidance* and *tension resolution*. Tension avoidance involves cleverly manipulating quorum sizes to avoid the tension entirely. This approach is used by Basic EPaxos [21] and Atlas [10]. The second technique, tension resolution, is significantly more complicated and involves detecting and resolving the tension through various means.

## 7 Tension Avoidance

In this section, we explain how to implement a generalized multi-leader state machine replication protocol using **tension avoidance**. The key idea behind tension avoidance is to avoid the tension between the consensus and dependency invariants

entirely. By manipulating quorum sizes in clever ways, we can ensure that whenever a proposer is forced to propose a set of dependencies $\text{deps}(v_x)$, this set of dependencies is guaranteed to satisfy the dependency invariant.

We first introduce Unanimous BPaxos, a simple protocol that implements tension avoidance. We then explain how Basic EPaxos and Atlas can be expressed as two optimized variants of Unanimous BPaxos.

### 7.1 Unanimous BPaxos

A Fast BPaxos deployment consists of $2f + 1$ servers. A proposer communicates with $f + 1$ acceptors in Phase 1 called a **Phase 1 quorum**, $f + \text{maj}(f + 1)$ acceptors in Phase 2 of round 0 called a **fast Phase 2 quorum**, and $f + 1$ acceptors in Phase 2 of rounds greater than 0 called a **classic Phase 2 quorum**. If we adjust the sizes of these quorums, we can avoid the tension between implementing consensus and computing dependencies. In [11], Howard et. al prove that Fast Paxos is safe so long as the following two conditions are met.

1. Every Phase 1 quorum and every classic Phase 2 quorum intersect. That is, for every Phase 1 quorum $Q$ and for every classic Phase 2 quorum $Q'$, $Q \cap Q' \neq \emptyset$.

2. Every Phase 1 quorum and every pair of fast Phase 2 quorums intersect. That is, for every Phase 1 quorum $Q$ and for every pair of fast Phase 2 quorum $Q', Q''$, $Q \cap Q' \cap Q'' \neq \emptyset$.

**Unanimous BPaxos** takes advantage of this result and increases the size of fast Phase 2 quorums. Specifically, Unanimous BPaxos is identical to Fast BPaxos except with fast Phase 2 quorums of size $2f + 1$. Unanimous BPaxos proposer pseudocode is given in Algorithm 3. It is identical to the pseudocode of a Fast Paxos proposer (Algorithm 1) except for a couple small changes highlighted in red.

**Algorithm 3** Unanimous BPaxos Proposer. Changes to Algorithm 1 are highlighted in red.

---

**State:** a value $v$, initially null
**State:** a round $i$, initially $-1$

 1: **upon** receiving PHASE2B$\langle 0, v' \rangle$ from all $2f + 1$ acceptors as the proposer of round 0 with $i = 0$ **do**
 2:     **if** every value of $v'$ is the same **then**
 3:         $v'$ is chosen, notify the learners
 4:     **else**
 5:         $i \leftarrow 1$
 6:         $v \leftarrow$ an arbitrary value satisfying the dependency invariant
 7:         send PHASE2A$\langle i, v \rangle$ to the acceptors
 8: **upon** recovery **do**
 9:     $i \leftarrow$ next largest round owned by this proposer
10:     send PHASE1A$\langle i \rangle$ to the acceptors
11: **upon** receiving PHASE1B$\langle i, vr, vv \rangle$ from $f + 1$ acceptors **do**
12:     $k \leftarrow$ the largest $vr$ in any PHASE1B$\langle i, vr, vv \rangle$
13:     **if** $k = -1$ **then**
14:         $v \leftarrow$ an arbitrary value satisfying the dependency invariant
15:     **else if** $k > 0$ **then**
16:         $v \leftarrow$ the corresponding $vv$ in round $k$
17:     **else if** $k = 0$ **then**
18:         **if** all $f + 1$ messages are of the form PHASE1B$\langle i, 0, v' \rangle$ for some value $v'$ **then**
19:             $v \leftarrow v'$
20:         **else**
21:             $v \leftarrow$ an arbitrary value satisfying the dependency invariant
22:     send PHASE2A$\langle i, v \rangle$ to the acceptors
23: **upon** receiving PHASE2B$\langle i \rangle$ from $f + 1$ acceptors **do**
24:     $v$ is chosen, notify the learners

---

Unlike Fast BPaxos, Unanimous BPaxos is safe. The critical change is on line 18. With fast Phase 2 quorums of size $2f + 1$ (line 1), a recovering proposer knows that a value $v'$ may have been chosen in round 0 only if all $f + 1$ acceptors that it communicates with in Phase 1 voted for $v'$ in round 0. If even a single acceptor did not vote for $v'$ in round 0, then $v'$ could not have received a unanimous vote and therefore was not chosen in round 0.

With Fast BPaxos, a proposer executing line 17 of Algorithm 1 is forced to propose a value $(x, \mathsf{deps}(v_x))$ if $\mathsf{maj}(f+1)$ acceptors voted for it in round 0, but the dependencies $\mathsf{deps}(v_x)$ may have only been computed by $\mathsf{maj}(f+1)$ dependency service nodes, violating the dependency invariant. This is exactly what happened in Figure 13. Unanimous BPaxos avoids the tension entirely because a proposer is only forced to propose a value $(x, \mathsf{deps}(v_x))$ if $f + 1$ acceptors voted for it in round 0. Now, we are guaranteed that $\mathsf{deps}(v_x)$ was computed

by a majority (i.e. $f + 1$) of the dependency service nodes. Thus, Unanimous BPaxos safely maintains the consensus and dependency service invariants.

The obvious disadvantage of Unanimous BPaxos is the protocol's large quorum sizes. In order to get a command chosen, a proposer has to perform a round trip of communication with *every* acceptor. This not only slows down the protocol in the normal case, it also decreases the protocol's ability to remain live in the face of faults. If even a single acceptor fails, the protocol grinds to a halt. This problem can be partially fixed by using more flexible quorums (like what Atlas [10] does) or by using a tension resolving protocol (see Section 7).

We now present two independent optimizations that improve the performance of Unanimous BPaxos. These optimizations were introduced in EPaxos [22] and Atlas [10].

## 7.2 Basic EPaxos Optimization

Unanimous BPaxos has a lower commit time than Simple BPaxos (4 network delays instead of 7), but has larger fast Phase 2 quorums ($2f + 1$ acceptors instead of $f + 1$). We now discuss an optimization, used by Basic EPaxos [22], to reduce the fast Phase 2 quorum size to $2f$.

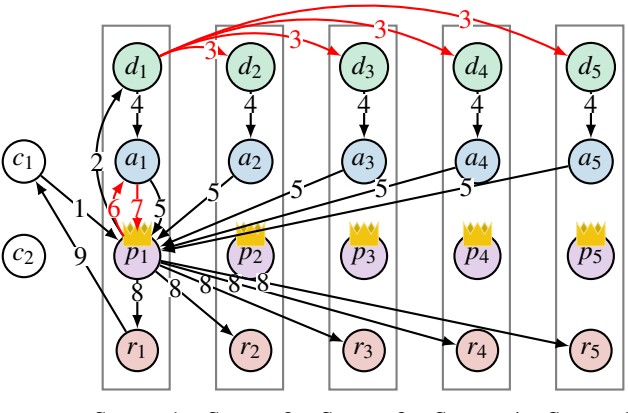

Figure 14: An example execution of Unanimous BPaxos ($f = 2$) with the Basic EPaxos optimization. The messages introduced by the optimization are drawn in red.

An execution of Unanimous BPaxos with the Basic EPaxos optimization is shown in Figure 14. We walk through the execution, highlighting the optimization's key changes. We assume $f > 1$ for now. Later, we discuss the case when $f = 1$.

- **(1)** When a client wants to propose a state machine command $x$, it sends $x$ to *any* of the proposers.

- **(2)** When a proposer $p_i$ receives a command $x$, from a client, it places $x$ in a vertex with globally unique vertex id $v_x = (p_i, m)$. **Change:** $p_i$ then sends $v_x$ and $x$ to the

co-located dependency service node $d_i$. It does not yet communicate with the other dependency service nodes.

- **(3) Change:** When $d_i$ receives $v_x$ and $x$, it computes the dependencies $\text{deps}(v_x)_i$ as usual using its acyclic conflict graph. $d_i$ then sends $v_x$, $x$, and $\text{deps}(v_x)_i$ to all the other dependency service nodes.

- **(4)** When a dependency service node $d_j$ receives $v_x$, $x$, and $\text{deps}(v_x)_i$, it computes the dependencies $\text{deps}(v_x)_j$ as usual using its acyclic conflict graph. **Change:** Then, $d_j$ proposes to its co-located acceptor $a_j$ that the value $(x, \text{deps}(v_x)_i \cup \text{deps}(v_x)_j)$ be chosen in vertex $v_x$ in round 0. $d_j$ combines the dependencies that it computed with the dependencies computed by $d_i$.

- **(5)** The acceptors are unchanged. In the normal case, when an acceptor $a_j$ receives value $(x, \text{deps}(v_x))$ in vertex $v_x = (p_i, m)$, it votes for the value and sends the vote to $p_i$.

- **(6) and (7) Change:** In Unanimous BPaxos, a value $v = (x, \text{deps}(v_x))$ is considered chosen in round 0 if all $2f + 1$ acceptors vote for $v$ in round 0. With the Basic EPaxos optimization, we only require $2f$ votes, and the act of choosing a value in round 0 is made more explicit. Specifically, if $p_i$ receives $2f$ votes for value $v = (x, \text{deps}(v_x))$ in round 0, including a vote from $a_i$, then it sends $v$ to the co-located acceptor $a_i$. When $a_i$ receives $v$ and is still in round 0 (i.e. $r = 0$ on Algorithm 2 line 0), then it records $v$ as chosen and responds to $p_i$. The value $v$ is considered chosen precisely when it is recorded by the acceptor. In the future $a_i$ responds to every PHASE1A and PHASE2A message with a notification that $v$ is chosen. If $a_i$ receives $v$ but is already in a round larger than 0 (i.e. $r > 0$), then it ignores $v$ and sends a negative acknowledgement back to $p_i$. Note that these messages are all performed locally on the server hosting $p_i$ and do not incur any network delay.

- **(8)** In the normal case, $p_i$ learns that $v$ was successfully chosen by $a_i$ and it broadcasts $v$ to all the acceptors. If $p_i$ receives a negative acknowledgement, it performs coordinated recovery as in Unanimous BPaxos.

- **(9)** The replicas are unchanged. They maintain and execute conflict graphs, returning the results of executing commands to the clients.

In addition to these changes made to the normal path of execution, the Basic EPaxos optimization also introduces a key change to the recovery procedure. Specifically, we replace line 18 – line 21 in Algorithm 3 with the following procedure.

Assume that proposer $p$ is recovering vertex $v_x = (p_j, m)$ in round $i > 0$. Either $p$ received a message from $a_j$ or it did not. We consider each case separately. First, assume that $p$ does receive a message from $a_j$. If $p$ receives a message indicating that some value $v'$ has already been chosen in round 0, then $p$ can terminate the recovery immediately. Otherwise, $p$ receives a PHASE1B message from $a_j$. From this, $p$ can conclude that $a_j$ is in a round at least as large as $i$ and therefore did not and will not record any value $v'$ chosen in round 0. Because of this, $p$ is safe to propose any value that satisfies the dependency invariant (e.g., $(\text{noop}, \emptyset)$).

Otherwise, $p$ does not receive a message from $a_j$. If $p$ receives $f$ PHASE1B$\langle i, 0, v' \rangle$ messages for the same value $v' = (x, \text{deps}(v_x))$, then $v'$ may have been chosen in round 0, so $p$ must propose $v'$ in order to maintain the consensus invariant. Note that $\text{deps}(v_x)$ also satisfies the dependency invariant despite the fact that $p$ only received $\text{deps}(v_x)$ from $f$, as opposed to $f + 1$, dependency service nodes. This is because the dependency service nodes that are not co-located with $d_j$ all propose dependencies that include the dependencies computed by $d_j$. Therefore, $p$ determines that $\text{deps}(v_x)$ is the union of $f + 1$ dependencies and maintains the dependency invariant. If $p$ does not receive $f$ PHASE1B$\langle i, 0, v' \rangle$ for the same value $v'$, then $p$ concludes no value was chosen or will be chosen in round 0, so $p$ is safe to propose any value that satisfies the dependency invariant.

Note that when $f = 1$ and $n = 3$, Phase 1 quorums, classic Phase 2 quorums, and fast Phase 2 quorums are all of size 2. This does *not* satisfy the conditions outlined by Howard et. al [11]. As a result, our protocol as stated is not safe for $f = 1$. The reason is that a recovering proposer may receive two different values in two separate PHASE1B messages from the two non-leader acceptors with values $v'$ and $v''$. In this situation, the proposer is unable to determine which value to propose. Thankfully, we can avoid this situation by having the leader send only to $2f$ acceptors rather than to all $2f + 1$ acceptors.

Ignoring some minor cosmetic differences, Unanimous BPaxos with the Basic EPaxos optimization is roughly equivalent to Basic EPaxos [22].

## 7.3 Atlas Optimization

In the best case, also called the fast path, Unanimous BPaxos achieves a commit time of four network delays. As shown in line 2 of Algorithm 3, a proposer executes the fast path only when every single acceptor votes for the exact same set of dependencies. As we saw in Figure 13, if any two dependency service nodes receive conflicting commands in different orders, their computed dependencies will not be the same. If a proposer does not receive a unanimous vote, it executes coordinated recovery, adding two more network delays to the commit time.

Atlas [10] introduces the following optimization to relax the requirement of a unanimous vote and increase the probability of a proposer executing the fast path. Let $X_1, \ldots, X_{2f+1}$ be $2f + 1$ sets. Let $\text{popular}(X_1, \ldots, X_{2f+1}) =$

$\{x \mid x$ appears in at least $f+1$ of the sets$\}$.

We change line 2 as follows. When a proposer receives dependencies $\text{deps}(v_x)_1, \ldots, \text{deps}(v_x)_{2f+1}$ from the $2f+1$ acceptors, it executes the fast path with dependencies $\text{deps}(v_x) = \text{deps}(v_x)_1 \cup \cdots \cup \text{deps}(v_x)_{2f+1}$ if $\text{deps}(v_x) = \text{popular}(\text{deps}(v_x)_1, \ldots, \text{deps}(v_x)_{2f+1})$. That is, the proposer takes the fast path only if every dependency $v_y$ computed by any of the dependency service nodes was computed by a majority of the dependency service nodes.

We also simplify line 18 – line 21. If a recovering proposer receives $f+1$ sets of dependencies, it proposes their union. Otherwise, it proposes an arbitrary value. This is safe because a set of dependencies $\text{deps}(v_x)$ can be chosen in round 0, only if every dependency in $\text{deps}(v_x)$ was computed by a majority of the dependency service nodes. Thus, every such element will appear in at least one of the $f+1$ dependency sets. Thus, the recovering proposer is sure to propose a dependency set if it was previously chosen (maintaining the consensus invariant), and it also proposes the union of $f+1$ dependency sets (maintaining the dependency invariant).

Atlas [10] is roughly equivalent to Unanimous BPaxos with the Basic EPaxos optimization, the Atlas optimization, and the flexible constraints on quorum sizes outlined in [11].

# 8 Tension Resolution

The advantage of tension avoidance is that it is simple. The disadvantage is that it requires large fast Phase 2 quorums. In this section, we explain how to implement a generalized multi-leader state machine replication protocol using **tension resolution**. Tension resolution is significantly more complicated than tension avoidance, but it does not require large fast Phase 2 quorums.

Instead of avoiding the tension between the consensus and dependency invariant, tension resolution uses additional machinery to resolve it when it arrives. Consider a scenario where a proposer $p$ is forced to propose a set of $\text{deps}(v_x)$ in round $i$ to maintain the consensus invariant because $\text{deps}(v_x)$ may have been chosen in a previous round. Simultaneously, $p$ is forced not to propose $\text{deps}(v_x)$ because it cannot guarantee that $\text{deps}(v_x)$ was computed by a majority of the dependency service nodes. This is the moment of tension that tension avoidance avoids. Tension resolution, on the other hand, allows this to happen. When it does, the proposer $p$ leverages additional machinery built into the protocol to determine either that (a) $\text{deps}(v_x)$ was not chosen or (b) $\text{deps}(v_x)$ was computed by a majority of dependency service nodes.

We introduce Majority Commit BPaxos, a protocol that implements tension resolution. We then discuss the protocol's relationship with EPaxos [22] and Caesar [10].

## 8.1 Pruned Dependencies

Before we discuss Majority Commit BPaxos, we introduce the notion of pruned dependencies. Imagine a proposer $p$ sends command $x$ to the dependency service in vertex $v_x$, and the dependency service computes the dependencies $\text{deps}(v_x)$. Let $v_y \in \text{deps}(v_x)$ be one of $v_x$'s dependencies. To maintain the dependency invariant, all of the protocols that we have discussed thus far would get $v_x$ chosen with a dependency on $v_y$, but this is not always necessary.

Assume that that the proposer $p$ knows that $v_y$ has been chosen with command $y$ and dependencies $\text{deps}(v_y)$. Further assume that $v_x \in \text{deps}(v_y)$. That is, $v_y$ has already been chosen with a dependency on $v_x$. In this case, there is no need for $v_x$ to depend on $v_y$. The dependency invariant asserts that if two vertices $v_a$ and $v_b$ are chosen with conflicting commands $a$ and $b$, then either $v_a \in \text{deps}(v_b)$ or $v_b \in \text{deps}(v_a)$. Thus, in our example, if $v_y$ has already been chosen with a dependency on $v_x$, then there is no need to propose $v_x$ with a dependency on $v_y$. Similarly, if $v_y$ has been chosen with noop as part of a recovery, then there is no need to propose $v_x$ with a dependency on $v_y$ because $x$ and noop do not conflict.

Let $\text{deps}(v_x)$ be a set of dependencies computed by the dependency service. Let $P \subseteq \text{deps}(v_x)$ be a set of vertices $v_y$ such that $v_y$ has been chosen with noop or $v_y$ has been chosen with $v_x \in \text{deps}(v_y)$. We call $\text{deps}(v_x) - P$ the **pruned dependencies** of $v_x$ with respect to $P$. Majority Commit BPaxos maintains Invariant 4, the **pruned dependency invariant**. The pruned dependency invariant is a relaxation of the dependency invariant. If a protocol maintains the pruned dependency invariant, it is guaranteed to maintain the dependency invariant.

**Invariant 4** (Pruned Dependency Invariant)**.** For every vertex $v_x$, either $(\text{noop}, \emptyset)$ is chosen in $v_x$ or $(x, \text{deps}(v_x) - P)$ is chosen in $v_x$ where $\text{deps}(v_x)$ are dependencies computed by the dependency service and where $\text{deps}(v_x) - P$ are the pruned dependencies of $v_x$ with respect to some set $P$.

## 8.2 Majority Commit BPaxos

For clarity of exposition, we first introduce a version of Majority Commit BPaxos that can sometimes deadlock. Later, we modify the protocol to eliminate the possibility of deadlock.

Majority Commit BPaxos is identical to Fast BPaxos except for the following two modifications. First, every Fast Paxos acceptor maintains a conflict graph in exactly the same way as the replicas do. That is, when an acceptor learns that a vertex $v_x$ has been chosen with command $x$ and dependencies $\text{deps}(v_x)$), it adds $v_x$ to its conflict graph with command $x$ and with edges to every vertex in $\text{deps}(v_x)$. We will see momentarily that whenever a Majority Commit BPaxos proposer sends a PHASE2A message to the acceptors with value $v = (x, \text{deps}(v_x) - P)$, the proposer also sends $P$ and all of the commands and dependencies chosen in in the vertices in

*P*. Thus, when an acceptor receives a PHASE2A message, it updates its conflict graph with the values chosen in *P*. Second and more substantially, a proposer executes a significantly more complex recovery procedure. This is shown in Algorithm 4.

As with Fast BPaxos, if $k = -1$ (line 3), if $k > 1$ (line 6), or if $k = 0$ and there does not exist $\mathsf{maj}(f+1)$ matching values (line 29), recovery is straightforward.

Otherwise, there does exist a $v' = (x, \mathsf{deps}(v_x))$ voted for by at least $\mathsf{maj}(f+1)$ acceptors in round 0 (line 9). As with Fast BPaxos, $v'$ may have been chosen in round 0, so the proposer *must* propose $v'$ in order to maintain the consensus invariant. But $\mathsf{deps}(v_x)$ may not be the union of dependencies computed by $f+1$ dependency service nodes, so the proposer is simultaneously forced *not* to propose $v'$ in order to maintain the dependency invariant. Unanimous BPaxos avoided this tension by increasing the size of fast Phase 2 quorums. Majority Commit BPaxos instead resolves the tension by performing a more sophisticated recovery procedure. In particular, the proposer does a bit of detective work to conclude either that $v'$ was definitely not chosen in round 0 (in which case, the proposer can propose a different value) or that $\mathsf{deps}(v_x)$ happens to be a pruned set of dependencies (in which case, proposer is safe to propose $v'$).

On line 11 and line 12, the proposer sends $v_x$ and $x$ to the dependency service nodes co-located with the acceptors in $A$ (i.e. the $f+1$ acceptors from which the proposer received PHASE1B messages). The proposer then computes the union of the returned dependencies, called $\mathsf{deps}(v_x)_A$. Note that this communication can be piggybacked on the PHASE1A messages that the proposer previously sent to avoid the extra round trip of communication. Also note that $\mathsf{deps}(v_x)$ was returned by $\mathsf{maj}(f+1)$ nodes in $A$, so $\mathsf{deps}(v_x)$ is a subset of $\mathsf{deps}(v_x)_A$.

Next, the proposer enters a for loop in an attempt to prune $\mathsf{deps}(v_x)_A$ until it is equal to $\mathsf{deps}(v_x)$. That is, the proposer attempts to construct a set of vertices $P$ such that $\mathsf{deps}(v_x) = \mathsf{deps}(v_x)_A - P$ is a set of pruned dependencies. For every, $v_y \in \mathsf{deps}(v_x)_A - \mathsf{deps}(v_x)$, the proposer first recovers $v_y$ if it does not know if a value has been chosen in vertex $v_y$ (line 17). After recovering $v_y$, assume the proposer learns that $v_y$ is chosen with command $y$ and dependencies $\mathsf{deps}(v_y)$. If $y = \mathsf{noop}$ or if $v_x \in \mathsf{deps}(v_y)$, then the proposer can safely prune $v_y$ from $\mathsf{deps}(v_x)_A$, so it adds $v_y$ to $P$ (line 19).

Otherwise, the proposer contacts some quorum $A'$ of acceptors (line 21). If any acceptor $a_j$ in $A'$ knows that vertex $v_x$ has already been chosen, then the proposer can abort the recovery of $v_x$ and retrieve the chosen value directly from $a_j$ (line 23). Otherwise, the proposer concludes that no value was chosen in $v_x$ in round 0 and is free to propose any value that maintains the dependency invariant (line 25). We will explain momentarily why the proposer is able to make such a conclusion. It is not obvious. Note that the proposer can piggyback its communication with $A'$ on its PHASE1A messages.

Finally, if the proposer exits the for loop, then it has successfully pruned $\mathsf{deps}(v_x)_A$ into $\mathsf{deps}(v_x)_A - P = \mathsf{deps}(v_x)$ and can safely propose it without violating the consensus or pruned dependency invariant (line 28). As described above, when the proposer sends a PHASE2A message with value $v'$, it also includes the values chosen in every vertex in $P$.

We now return to line 25 and explain how the proposer is able to conclude that $v'$ was not chosen in round 0. On line 25, the proposer has already concluded that $v_y$ was not chosen with noop and that $v_x \notin \mathsf{deps}(v_y)$. By the pruned dependency invariant, $\mathsf{deps}(v_y) = \mathsf{deps}(v_y)_D - P'$ is a set of pruned dependencies where $\mathsf{deps}(v_y)_D$ is a set of dependencies computed by a set $D$ of $f+1$ dependency service nodes. Because $v_x \notin \mathsf{deps}(v_y)_D - P'$, either $v_x \notin \mathsf{deps}(v_y)_D$ or $v_x \in P'$.

$v_x$ cannot be in $P'$ because if $v_y$ were chosen with dependencies $\mathsf{deps}(v_y)_D - P'$, then some quorum of acceptors would have received $P'$ and learned that $v_x$ was chosen. But, when the proposer contacted the quorum $A'$ of acceptors, none knew that $v_x$ was chosen, and any two quorums intersect.

Thus, $v_x \notin \mathsf{deps}(v_y)_D$. Thus, every dependency service node in $D$ processed instance $v_y$ before instance $v_x$. If not, then a dependency service node in $D$ would have computed $v_x$ as a dependency of $v_y$. However, if every dependency service node in $D$ processed $v_y$ before $v_x$, then there cannot exist a fast Phase 2 quorum of dependency service nodes that processed $v_x$ before $v_y$. In this case, $v' = (x, \mathsf{deps}(v_x))$ could not have been chosen in round 0 because it necessitates a fast Phase 2 quorum of dependency service nodes processing $v_x$ before $v_y$ because $v_y \notin \mathsf{deps}(v_x)$.

## 8.3   Ensuring Liveness

Majority Commit BPaxos is safe, but it is not very live. There are certain failure-free situations in which Majority Commit BPaxos can permanently deadlock. The reason for this is line 17 in which a proposer defers the recovery of one vertex for the recovery of another. There exist executions of Majority Commit BPaxos with a chain of vertices $v_1, \ldots, v_m$ where the recovery of every vertex $v_i$ depends on the recovery of vertex $v_{i+1 \bmod m}$.

We now modify Majority Commit BPaxos to prevent deadlock. First, we change the condition under which a value is considered chosen on the fast path. A proposer considers a value $v = (x, \mathsf{deps}(v_x))$ chosen on the fast path if a fast Phase 2 quorum $F$ of acceptors voted for $v$ in round 0 *and* for every vertex $v_y \in \mathsf{deps}(v_x)$, there exists a quorum $A \subseteq F$ of $f+1$ acceptors that knew $v_y$ was chosen at the time of voting for $v$. Second, when an acceptor $a_i$ sends a PHASE2B vote in round 0 for value $v = (x, \mathsf{deps}(v_x))$, $a_i$ also includes the subset of vertices in $\mathsf{deps}(v_x)$ that $a_i$ knows are chosen, as well as the values chosen in these vertices. Third, proposers execute Algorithm 4 but with the lines of code shown in Algorithm 5 inserted after line 10.

We now explain Algorithm 5. On line 11, the proposer

**Algorithm 4** Majority Commit BPaxos Proposer. Pseudocode for initiating recovery and handling PHASE2B messages is ommitted because it is identical to the pseudocode in Algorithm 1.

---

**State:** a value $v$, initially null
**State:** a round $i$, initially $-1$
1: **upon** receiving PHASE1B$\langle i, vr, vv \rangle$ from $f + 1$ acceptors $A$ **do**
2:     $k \leftarrow$ the largest $vr$ in any PHASE1B$\langle i, vr, vv \rangle$
3:     **if** $k = -1$ **then**
4:         $v \leftarrow$ an arbitrary value satisfying the dependency invariant
5:         send PHASE2A$\langle i, v \rangle$ to the acceptors
6:     **else if** $k > 0$ **then**
7:         $v \leftarrow$ the corresponding $vv$ in round $k$
8:         send PHASE2A$\langle i, v \rangle$ to the acceptors
9:     **else if** there are maj$(f + 1)$ PHASE1B$\langle i, 0, v' \rangle$ messages for some value $v'$ **then**
10:         $(x, \text{deps}(v_x)) \leftarrow v'$
11:         send $v_x$ and $x$ to the dependency service nodes co-located with the acceptors in $A$
12:         $\text{deps}(v_x)_A \leftarrow$ the union of the dependencies returned by these dependency service nodes
13:
14:         $P \leftarrow \emptyset$
15:         **for** $v_y \in \text{deps}(v_x)_A - \text{deps}(v_x)$ **do**
16:             **if** we don't know if $v_y$ is chosen **then**
17:                 recover $v_y$, blocking until $v_y$ is recovered
18:             **if** $v_y$ chosen with noop or with $v_x \in \text{deps}(v_y)$ **then**
19:                 $P \leftarrow P \cup \{v_y\}$
20:             **else**
21:                 contact a quorum $A'$ of acceptors
22:                 **if** an acceptor in $A'$ knows $v_x$ is chosen **then**
23:                     abort recovery; $v_x$ has already been chosen
24:                 **else**
25:                     $v \leftarrow$ an arbitrary value satisfying the dependency invariant
26:                     send PHASE2A$\langle i, v \rangle$ to the acceptors
27:         $v \leftarrow v'$
28:         send PHASE2A$\langle i, v \rangle$ and the values chosen in $P$ to at least $f + 1$ acceptors
29:     **else**
30:         $v \leftarrow$ an arbitrary value satisfying the dependency invariant
31:         send PHASE2A$\langle i, v \rangle$ to the acceptors

---

computes the subset $M \subseteq A$ of acceptors that voted for $v'$ in round 0. On line 12, the proposer determines whether there exists some instance $v_y \in \text{deps}(v_x)$ such that no acceptor in $M$ knows that $v_y$ is chosen. If such an $v_y$ exists, then $v'$ was not chosen in round 0. To see why, assume for contradiction that $v'$ was chosen in round 0. Then, there exists some fast Phase 2 quorum $F$ of acceptors that voted for $v'$ in round 0, and there exists some quorum $A' \subseteq F$ of acceptors that know $v_y$ has been chosen. However, $A$ and $A'$ intersect, but no acceptor in $A$ both voted for $v'$ in round 0 and knows that $v_y$ was chosen. This is a contradiction. Thus, the proposer is free to propose any value satisfying the dependency invariant.

Next, it's possible that the proposer was previously recovering instance $v_z$ with value $(z, \text{deps}(v_z))$ and executed line 17 of Algorithm 4, deferring the recovery of instance $v_z$ until after the recovery of instance $v_x$. If so and if $v_z \in \text{deps}(v_x)$,

then some acceptor $a_j \in M$ knows that $v_z$ is chosen. Thus, the proposer can abort the recovery of instance $v_z$ and retrieve the chosen value directly from $a_j$ (line 16). Otherwise, $v_z \notin \text{deps}(v_x)$. In this case, no value was chosen in round 0 of instance $v_z$, so the proposer is free to propose any value satisfying the pruned dependency invariant in instance $v_z$. Here's why. $v_z \notin \text{deps}(v_x)$, so every dependency service node co-located with an acceptor in $M$ processed $v_x$ before $v_z$. $|M| \geq \text{maj}(f + 1)$, so there strictly fewer than $f + \text{maj}(f + 1)$ remaining dependency service nodes that could have processed $v_z$ before $v_x$. If the proposer was recovering instance $v_z$ but deferred to the recovery of instance $v_x$, then $v_x \notin \text{deps}(v_z)$. In order for $v_z$ to have been chosen in round 0 with $v_x \notin \text{deps}(v_y)$, it requires that at least $f + \text{maj}(f)$ dependency service nodes processed $v_z$ before $v_x$, which we just concluded is impossible. Thus, $v_z$ was not chosen in

**Algorithm 5** Majority Commit BPaxos proposer modification to prevent deadlock.

---

11:  $M \leftarrow$ the set of acceptors in $A$ that voted for $v'$ in round 0
12:  **if** $\exists v_y \in \mathrm{deps}(v_x)$ such that no acceptor in $M$ knows
         that $v_y$ is chosen **then**
13:     send any value satisfying the dependency invariant
14:  **if** the proposer was recovering $v_z$ and deferred to the
        recovery of $v_x$ **then**
15:     **if** $v_z \in \mathrm{deps}(v_x)$ **then**
16:        abort recovery of $v_z$; $v_z$ has already been chosen
17:     **else**
18:        in vertex $v_z$, send any value satisfying the
               dependency invariant

---

round 0.

Majority Commit BPaxos is deadlock free for the following reason. If a proposer is recovering instance $v_z$ and defers to the recovery of instance $v_x$, then either the proposer will recover $v_x$ using line 12 of Algorithm 5 or the proposer will recover $v_z$ using line 16 or line 18 of Algorithm 5. In either case, any potential deadlock is avoided.

### 8.4   EPaxos and Caesar

EPaxos [22] and Caesar [3] are two generalized multi-leader protocols that implement tension resolution. EPaxos is very similar Majority Commit BPaxos with the Basic EPaxos optimization from Section 7.2 used to reduce fast Phase 2 quorum sizes by 1. Majority Commit BPaxos and EPaxos both prune dependencies and perform a recursive recovery procedure with extra machinery to avoid deadlocks. Caesar improves on EPaxos in two dimensions. First, much like Atlas, a Caesar proposer does not require that a fast Phase 2 quorum of acceptors vote for the exact same value in order to take the fast path. Second, Caesar avoids a recursive recovery procedure. Caesar accomplishes this using a combination of logical timestamps and carefully placed barriers in the protocol.

## 9   Related Work

**MultiPaxos, Raft, Viewstamped Replication**  Generalized multi-leader protocols have a number of advantages over single leader protocols like MultiPaxos [16], Raft [23], and Viewstamped Replication [18] that totally order commands into a log. See [22] for more details and experimental validation.

First, generalized multi-leader protocols avoid being bottlenecked by a single leader. In protocols like MultiPaxos and Raft, all state machine commands are funneled through a single leader, making the leader the throughput bottleneck. In multi-leader protocols on the other hand, state machine commands can be processed by *any* of the multiple leaders,

preventing any one leader from becoming a bottleneck. This allows multi-leader protocols to achieve higher throughput.

Second, generalized multi-leader protocols like EPaxos are more resilient to leader failures. With protocols like Multi-Paxos and Raft, when the leader fails, the protocol's throughput drops to zero and stays at zero until the failure is detected and a new leader is elected. Depending on the deployment, this delay could be seconds or minutes. With protocols like EPaxos on the other hand, when a leader fails, the protocol's throughput drops, but not to zero. All other non-failed leaders can still process commands, so the throughput remains high. When the failed leader is replaced, the throughput returns to normal.

Third, generalized multi-leader protocols achieve lower latency in geo-distributed applications. Consider a geo-replicated deployment of MultiPaxos. If the MultiPaxos leader is in Europe, the clients in North America will experience much higher latency than the clients in Europe. In general, the clients that are geographically close to the single leader will experience low latency, while all other clients will experience significantly higher latency. With generalized multi-leader protocols, the multiple leaders can be distributed across the deployment so that every client has a leader that is geographically close by. This reduces the overall latency of the protocol.

Fourth, generalized multi-leader protocols have lower tail latencies for applications with little interdependence between commands. With protocols like MultiPaxos, if a single log entry is delayed (e.g., because of a network failure), all subsequent commands in the log are also delayed. Thus, any slowdown in the execution of a single command can affect many commands serialized after it. With generalized multi-leader protocols, independent commands are executed independently and do not wait for each other. Thus, if a single command is slow to execute, the other independent commands are not affected.

**A Family of Leaderless Generalized Consensus Algorithms**  In [19], Losa et al. propose a generic generalized consensus algorithm that is structured as the composition of a generic dependency-set algorithm and a generic map-agreement algorithm. The invariants of the dependency-set and map-agreement algorithm are very similar to the consensus and dependency invariants, and the example implementations of these two algorithms in [19] form an algorithm similar to Simple BPaxos. Our paper builds on this body of work by introducing Fast BPaxos, Unanimous BPaxos, and Majority Commit BPaxos. We also identify the tension between the two invariants as the key distinguishing feature of most protocols and taxonimize existing protocols by how they handle the tension.

**Generalized Paxos and GPaxos**  Generalized Paxos [14] and GPaxos [26] are generalized protocols but are not fully

multi-leader. Clients send commands directly to acceptors, behaving very much like a leader. However, in the face of collisions, Generalized Paxos and GPaxos rely on a single leader to resolve the collision. This single leader becomes a bottleneck in high contention workloads and prevents scaling.

**SpecPaxos, NOPaxos, CURP**  SpecPaxos [25] and NOPaxos [17] combine speculative execution and ideas from Fast Paxos in order to reduce commit delay as low as two network delays. CURP [24] further introduces generalization, allowing commuting commands to be executed in any order. These protocols represent yet another point in the design space of state machine replication protocols. As the commit delay decreases, the complexity of the protocols generally increases. We think this is an exciting avenue of research and hope that an improved understanding of generalized multi-leader protocols can accelerate research in this direction.

**Mencius**  Mencius [20] is a multi-leader, non-generalized protocol in which MultiPaxos log entries are round-robin partitioned among a set of leaders. Because Mencius is not generalized, a log entry cannot be executed until *all* previous log entries have been executed. To ensure log entries are being filled in appropriately, Mencius leaders perform all-to-all communication between each other. Mencius is significantly less complex that generalized multi-leader protocols like EPaxos, Caesar, and Atlas. This demonstrates that much of the complexity of these protocols come from being generalized rather than being multi-leader, though both play a role.

**Chain Replication**  Chain Replication [31] is a state machine replication protocol in which the set of state machine replicas are arranged in a totally ordered chain. Writes are propagated through the chain from head to tail, and reads are serviced exclusively by the tail. Chain Replication has high throughput compared to MultiPaxos because load is more evenly distributed between the replicas. This shows that the leader bottleneck can be addressed without necessarily having multiple leaders.

**Scalog**  Scalog [9] is a replicated shared log protocol that achieves high throughput using a sophisticated form of batching. A client does not send values directly to a centralized leader for sequencing in the log. Instead, the client sends its values to one of a number of batchers. Periodically, the batchers' batches are sealed and assigned an id. This id is then sent to a state machine replication protocol, like MultiPaxos, for sequencing. Like Mencius, Scalog represents a way to avoid a leader bottleneck without needing multiple leaders.

**PQR, Harmonia, and CRAQ**  PQR [7], Harmonia [32], and CRAQ [28] all implement optimizations so that reads (i.e. state machine commands that do not modify the state of the state machine) can be executed without contacting a leader, while writes are still processed by the leader. An interesting direction of future work could explore whether or not these read optimizations could be applied to generalized multi-leader protocols.

## 10   Conclusion

In this paper, we explained, analyzed, and taxonomized generalized multi-leader state machine replication protocols. Our taxonomy of state machine replication protocols is summarized in Figure 15, and a summary of the generalized multi-leader protocols that we discuss in this paper is given in Table 2. We showed via Simple BPaxos that simple generalized multi-leader protocols do exist, but they have high commit time. Reducing the commit time with Fast BPaxos, we discovered the fundamental tension between implementing consensus and computing dependencies between commands. We taxonomized existing protocols according to whether they avoid the tension (like Unanimous BPaxos) or they resolve the tension (like Majority Commit BPaxos). Ultimately, we hope that the clarity we have brought to the space can encourage more industry adoption of generalized multi-leader protocols and can spur new academic innovations in this space.

## Acknowledgement

This research is supported in part by DHS Award HSHQDC-16-3-00083, NSF CISE Expeditions Award CCF-1139158, and gifts from Alibaba, Amazon Web Services, Ant Financial, CapitalOne, Ericsson, GE, Google, Huawei, Intel, IBM, Microsoft, Scotiabank, Splunk and VMware.

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

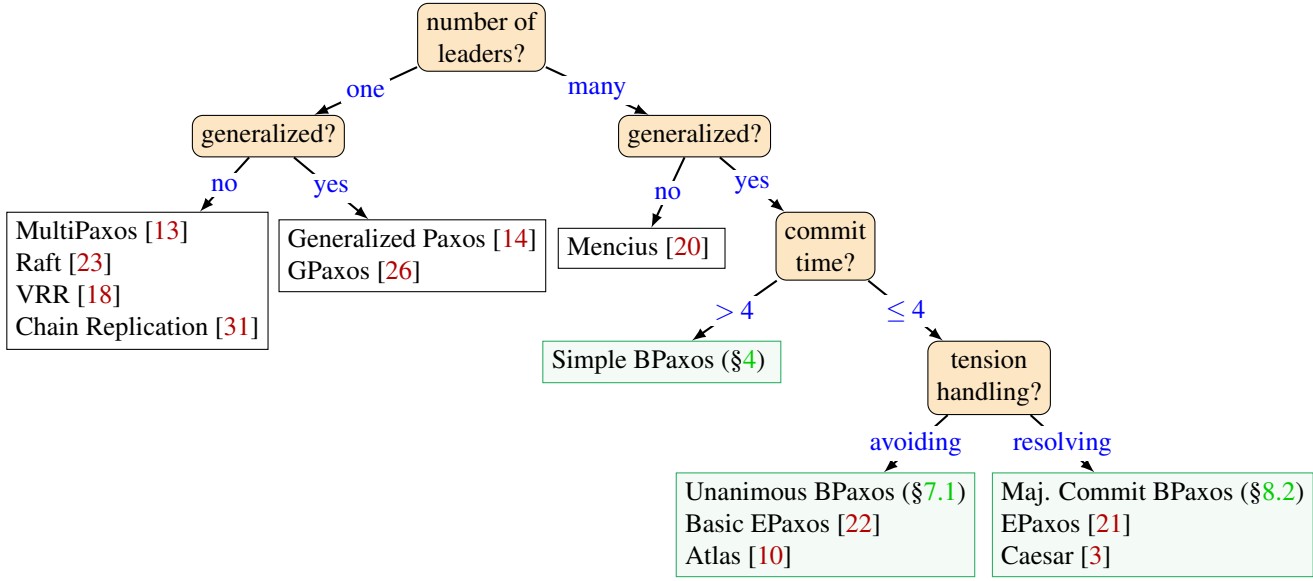

Figure 15: A non-exhaustive taxonomy of state machine replication protocols. The generalized multi-leader protocols that we discuss in this paper are shaded green.

Table 2: A summary of generalized multi-leader state machine replication protocols.

| Protocol | Safe | Commit Time | Tension Handling | Number of Nodes | Phase 1 Quorum Size | Classic Phase 2 Quorum Size | Fast Phase 2 Quorum Size |
|---|---|---|---|---|---|---|---|
| Simple BPaxos (§4) | yes | 7 | N/A | $2f+1$ | $f+1$ | $f+1$ | N/A |
| Fast BPaxos (§6) | no | 4 | N/A | $2f+1$ | $f+1$ | $f+1$ | $f+\mathsf{maj}(f+1)$ |
| Unanimous BPaxos (§7.1) | yes | 4 | avoidance | $2f+1$ | $f+1$ | $f+1$ | $2f+1$ |
| Basic EPaxos [22] | yes | 4 | avoidance | $2f+1$ | $f+1$ | $f+1$ | $2f$ |
| Atlas [10] | yes | 4 | avoidance | $n$ | $f+1$ | $n-f$ | $\lfloor\frac{n}{2}\rfloor+f$ |
| Maj. Commit BPaxos (§8.2) | yes | 4 | resolution | $2f+1$ | $f+1$ | $f+1$ | $f+\mathsf{maj}(f+1)$ |
| EPaxos [21] | yes | 4 | resolution | $2f+1$ | $f+1$ | $f+1$ | $f+\mathsf{maj}(f+1)-1$ |
| Caesar [3] | yes | 4 | resolution | $2f+1$ | $f+1$ | $f+1$ | $f+\mathsf{maj}(f+1)$ |

[4] Jason Baker, Chris Bond, James C Corbett, JJ Furman, Andrey Khorlin, James Larson, Jean-Michel Leon, Yawei Li, Alexander Lloyd, and Vadim Yushprakh. Megastore: Providing scalable, highly available storage for interactive services. In *CIDR*, volume 11, pages 223–234, 2011.

[5] Mike Burrows. The chubby lock service for loosely-coupled distributed systems. In *Proceedings of the 7th symposium on Operating systems design and implementation*, pages 335–350. USENIX Association, 2006.

[6] Tushar D Chandra, Robert Griesemer, and Joshua Redstone. Paxos made live: an engineering perspective. In *Proceedings of the twenty-sixth annual ACM symposium on Principles of distributed computing*, pages 398–407. ACM, 2007.

[7] Aleksey Charapko, Ailidani Ailijiang, and Murat Demirbas. Linearizable quorum reads in paxos. In *11th USENIX Workshop on Hot Topics in Storage and File Systems (HotStorage 19)*, 2019.

[8] James C Corbett, Jeffrey Dean, Michael Epstein, Andrew Fikes, Christopher Frost, Jeffrey John Furman, Sanjay Ghemawat, Andrey Gubarev, Christopher Heiser, Peter Hochschild, et al. Spanner: Google's globally distributed database. *ACM Transactions on Computer Systems (TOCS)*, 31(3):8, 2013.

[9] Cong Ding, David Chu, Evan Zhao, Xiang Li, Lorenzo Alvisi, and Robbert van Renesse. Scalog: Seamless reconfiguration and total order in a scalable shared log. In *17th USENIX Symposium on Networked Systems Design and Implementation (NSDI 20)*, pages 325–338, 2020.

[10] Vitor Enes, Carlos Baquero, Tuanir França Rezende, Alexey Gotsman, Matthieu Perrin, and Pierre Sutra. State-machine replication for planet-scale systems. In

*Proceedings of the Fifteenth European Conference on Computer Systems*, pages 1–15, 2020.

[11] Heidi Howard, Aleksey Charapko, and Richard Mortier. Fast flexible paxos: Relaxing quorum intersection for fast paxos. In *International Conference on Distributed Computing and Networking 2021*, pages 186–190, 2021.

[12] Heidi Howard and Richard Mortier. Paxos vs raft: Have we reached consensus on distributed consensus? In *Proceedings of the 7th Workshop on Principles and Practice of Consistency for Distributed Data*, pages 1–9, 2020.

[13] Leslie Lamport. The part-time parliament. *ACM Transactions on Computer Systems (TOCS)*, 16(2):133–169, 1998.

[14] Leslie Lamport. Generalized consensus and paxos. 2005.

[15] Leslie Lamport. Fast paxos. *Distributed Computing*, 19(2):79–103, 2006.

[16] Leslie Lamport et al. Paxos made simple. *ACM Sigact News*, 32(4):18–25, 2001.

[17] Jialin Li, Ellis Michael, Naveen Kr Sharma, Adriana Szekeres, and Dan RK Ports. Just say NO to paxos overhead: Replacing consensus with network ordering. In *12th USENIX Symposium on Operating Systems Design and Implementation (OSDI 16)*, pages 467–483, 2016.

[18] Barbara Liskov and James Cowling. Viewstamped replication revisited. 2012.

[19] Giuliano Losa, Sebastiano Peluso, and Binoy Ravindran. Brief announcement: A family of leaderless generalized-consensus algorithms. In *Proceedings of the 2016 ACM Symposium on Principles of Distributed Computing*, pages 345–347. ACM, 2016.

[20] Yanhua Mao, Flavio P Junqueira, and Keith Marzullo. Mencius: building efficient replicated state machines for wans. In *8th USENIX Symposium on Operating Systems Design and Implementation (OSDI 08)*, pages 369–384, 2008.

[21] Iulian Moraru, David G Andersen, and Michael Kaminsky. A proof of correctness for egalitarian paxos. Technical report, Technical report, Parallel Data Laboratory, Carnegie Mellon University, 2013.

[22] Iulian Moraru, David G Andersen, and Michael Kaminsky. There is more consensus in egalitarian parliaments. In *Proceedings of the Twenty-Fourth ACM Symposium on Operating Systems Principles*, pages 358–372. ACM, 2013.

[23] Diego Ongaro and John K Ousterhout. In search of an understandable consensus algorithm. In *USENIX Annual Technical Conference*, pages 305–319, 2014.

[24] Seo Jin Park and John Ousterhout. Exploiting commutativity for practical fast replication. In *16th USENIX Symposium on Networked Systems Design and Implementation (NSDI 19)*, pages 47–64, 2019.

[25] Dan RK Ports, Jialin Li, Vincent Liu, Naveen Kr Sharma, and Arvind Krishnamurthy. Designing distributed systems using approximate synchrony in data center networks. In *NSDI*, pages 43–57, 2015.

[26] Pierre Sutra and Marc Shapiro. Fast genuine generalized consensus. In *Reliable Distributed Systems (SRDS), 2011 30th IEEE Symposium on*, pages 255–264. IEEE, 2011.

[27] Rebecca Taft, Irfan Sharif, Andrei Matei, Nathan VanBenschoten, Jordan Lewis, Tobias Grieger, Kai Niemi, Andy Woods, Anne Birzin, Raphael Poss, Paul Bardea, Amruta Ranade, Ben Darnell, Bram Gruneir, Justin Jaffray, Lucy Zhang, and Peter Mattis. Cockroachdb: The resilient geo-distributed sql database. In *Proceedings of the 2020 ACM SIGMOD International Conference on Management of Data*, pages 1493–1509. ACM, 2020.

[28] Jeff Terrace and Michael J Freedman. Object storage on craq: High-throughput chain replication for read-mostly workloads. In *USENIX Annual Technical Conference*, number June, pages 1–16. San Diego, CA, 2009.

[29] Alexander Thomson, Thaddeus Diamond, Shu-Chun Weng, Kun Ren, Philip Shao, and Daniel J Abadi. Calvin: fast distributed transactions for partitioned database systems. In *Proceedings of the 2012 ACM SIGMOD International Conference on Management of Data*, pages 1–12. ACM, 2012.

[30] Robbert Van Renesse and Deniz Altinbuken. Paxos made moderately complex. *ACM Computing Surveys (CSUR)*, 47(3):42, 2015.

[31] Robbert Van Renesse and Fred B Schneider. Chain replication for supporting high throughput and availability. In *OSDI*, volume 4, 2004.

[32] Hang Zhu, Zhihao Bai, Jialin Li, Ellis Michael, Dan RK Ports, Ion Stoica, and Xin Jin. Harmonia: Near-linear scalability for replicated storage with in-network conflict detection. *Proceedings of the VLDB Endowment*, 13(3):376–389, 2019.

