# OpenReview forum: "SoK: A Generalized Multi-Leader State Machine Replication Tutorial"
_JSYS/2021/Mar_Papers — JSYS Mar 21_

### Official Review · AnonReviewer2 · 2021-04-01
**Very approachable paper on multi-leader consensus protocols but lacks elaboration on read-only operations**

**Decision:**

Weak accept: good paper with flaws that can be fixed in three months

**Review:**

Authors set a goal to encourage an industry adoption of the multi-leader protocols so I look and evaluate the paper from a position of a practitioner trying to build a multi-leader consensus based system. The explanation of fast paxos and multi-leader protocols is clear and it helped me to develop an intuition for the protocols sufficient for building a prototype.

---------

Overall it is a good paper: it not only refines the explanations but also introduces an abstraction helping to reason about the multi-leader consensus protocols (Simple PPaxos) and describes a new optimization / multi leader protocol (Majority Commit PPaxos). But the paper lacks several important characteristics necessary for building a production system:

  * a reconfiguration routine
  * information on the retrieving data from the state machines
  * measurable benefits of switching to from Raft/MultiPaxos to the multi-leader protocols

## Measurable benefits of switching to multi-leader protocols

Multi-leader consensus protocols are complex and it's harder to implement them than their leadered counterparts but in theory they have better performance. As a practitioner I wonder if the benefits are worth the complexity but the paper doesn't give me an answer.

The authors advocate that the major benefit comes as increased throughput (in Raft/Paxos a leader sends and receives disproportionately more messages). I recommend to include either experiment data or an estimation of the upper limits of throughput for both systems.

E.g. napkin math shows that the throughput of multi-leader consensus is `(2f+1)^2/(4f+1)` times better than leadered where `f` is max allowed numbers of failures, proposer, acceptors and learners are colocated, it uses `f+1` sized quorums and the command's payload is way more bigger than the result of command execution.

## Retrieving data from the state machines

Usually production Paxos-based systems depend on bypassing the replication protocol for serving reads but the paper doesn't contain any information on how to do it. For example the leader-based protocols have well understood models for lightweight reads:

  - to read from a leader after waiting for a round of the heartbeats or after checking that its lease doesn't expire
  - to read from a combination of the followers using methods from "Linearizable Quorum Reads in Paxos" or "Paxos quorum leases: Fast reads without sacrificing writes" papers
  - to accept a possibility of staleness and to read from any node

But those models aren't applicable to the multi-leader case: it lacks the leader/follower roles and local reads lead not only to staleness but to observing incompatible histories too caused by the reordering of the commuting commands.

## Reconfiguration

The last important missing part is reconfiguration. It's crucial from a practical perspective. Without reconfiguration there is no way to replace the failed nodes and since failures are inevitable then eventually the multi-leader consensus based systems lose a majority and become unavailable.

One of the reasons Raft got so widespread in the industry is an included reconfiguration protocol (joined consensus) so with the declared goal I recommend to update the paper to include the reconfiguration sub-protocol.

-----------------

## Other (minor) issues

"This paper assumes that at most f machines will fail" - from a practical standpoint it's impossible to guarantee; it's better to mention what happens then this threshold is passed if consistency or availability becomes violated.

"To reach is consensus on a value, an execution of Paxos is divided into a number of integer value rounds" - it's a clever idea to pre-assign a set of ballot numbers to the machines but using a term 'round' both for a ballot number and an act of communication is confusing. It may create a false impression that it's impossible to skip ballot numbers. Maybe it'd be better to mention that rounds also known as ballot numbers (paxos) or term (raft) and add a reference to "Paxos vs Raft: Have we reached consensus on distributed consensus?" to use it as a map between different protocols.

**Expertise:**

Follow the literature closely, last published 5+ years ago

**Useful:**

yes

---

### Official Review · AnonReviewer4 · 2021-04-08
**Great tutorial paper on multi-leader Paxos**

**Decision:**

Weak accept: good paper with flaws that can be fixed in three months

**Review:**

I enjoyed reading this tutorial paper on generalized multi-leader state machine replication. The paper gives a very simple protocol that nicely captures the essence of such protocols. The paper then presents an unsafe protocol to set the stage for why advanced protocols work the way they do in the literature. I am usually skeptical about deriving insights from unsafe protocols, but this paper does a good job with this approach.

However, I am not entirely certain what went wrong for Fast PPaxos. The paper claims that it is due to a dilemma between reaching consensus and preserving dependency. But it seems to me that the bug is a result of multi-leader rather than dependency. It seems that the bug could be avoided by simply abandoning concurrent leaders. The first approach to fix this bug, increasing quorum size to 2f+1, also does not seem related to dependency; rather, it seems to be avoiding a tie between concurrent leaders.  I would like to see more discussion on the root cause of the problem, and how exactly it is fixed. This is very important for the paper since at some point the authors claim that most of the design complexity comes from generalized as opposed to multi-leader.

Other than that, I have a few minor suggestions in terms of presentation.
- When describing Fast Paxos, the authors use proposals, acceptors, and learners; but in Simple or Fast PPaxos, the authors use proposals, acceptors, and replicas. I suggest the authors stick to one convention, preferrably learners.

- Page 11 right column touched on why the threshold must be > 1.5f. While this may be obvious for experts, it is helpful to elaborate on this in a tutorial paper for non-expert readers.

- Section 2 says that messages can be dropped, and indeed Section 6.3  considers a situation that messages to d3, d4, d5 are dropped. However, the default model of Paxos does not allow message drop. Otherwise, one runs into the two-general impossibility. In this particular example in Section 6.3, can we simply have p1 crash before it sends messages to d3, d4,and d5? If this is sufficient to break Fast PPaxos, then I suggest removing message drop from the model.  If this is not sufficient, the authors need to carefully define the model regarding message drop, and explain why the two general's impossibility does not apply.


Overall, I like this tutorial paper and would support its acceptance as a tutorial paper. However, it is currently submitted as an SoK paper.  If judged as an SoK paper, I am less supportive. The paper mentions that there are few generalized multi-leader systems, and indeed it only reviews 3 papers in detail (EPaxos, Caesar, Atlas). The related work section mentions a few other works very briefly and they are not closely related.  As such, I feel the area of generalized multi-leader state machine replication is still relatively young and small, and has not reached a stage of needing an SoK paper. This is not a criticism of the paper. I think the authors are well aware that their paper is more of a tutorial than SoK. The onus is on the editorial board to accept this paper under an appropriate category.








**Expertise:**

Follow the literature closely, last published 5+ years ago

**Useful:**

yes

---

### Official Review · AnonReviewer3 · 2021-04-10
**Review for paper titled, "SoK: A Generalized Multi-Leader State Machine Replication Tutorial"**

**Decision:**

Weak accept: good paper with flaws that can be fixed in three months

**Review:**

## Summary:
This paper aims to explain the space of multi-leader consensus protocols. The key advantage of adopting multi-leader protocols is that they facilitate out-of-order execution of commutating. In specific, if two commands do not conflict, then these protocols allow replicas to execute them in their desired order. To explain this space, the paper starts with a primer on SMR and Paxos, following which it explains MultiPaxos and defines the notion of conflict graphs. These conflict graphs are essential to the design of these multi-leader protocols as they specify the commands that can commute. To explain how existing multi-leader protocols work, authors lay down two key invariants: consensus invariant and dependency invariant. Next, authors gradually lay down the design of existing multi-leader protocols by presenting four of their variants: Simple PPaxos, Fast PPaxos, Unanimous PPaxos, and Majority Commit PPaxos. The paper also shows how these variants can be mapped to existing protocols like Basic Epaxos, Atlas, and so on.

## Pros:
1. I think this paper is a timely addition to this space and has potential to become a paper like Raft that cleanly explains the domain of multi-leader protocols. With so many multi-leader protocols, a survey paper like this can be useful for both pedagogical and engineering purposes.
2. The first seven pages of this paper are really fun to read and insightful. The authors have laid down the framework very well and kudos to them for the nice description. To motivate the spectrum, authors have started by explaining even Paxos, and as a reader, this really helps set up the pace.
3. I found the paper as a whole to be very descriptive. For each protocol, authors have presented the algorithm, an accompanying figure, and an example. This helps inn visualizing the problem at hand. Further, the story is finely knit. Authors explain why Simple PPaxos is inefficient and then show how it can be improved by Fast PPaxos. However, Fast PPaxos is unsafe, so there are two approaches to follow: Unanimous PPaxos or Majority Commit PPaxos.

## Cons:
1. I strongly believe that this paper is missing some evaluation. It will be really helpful if the authors can illustrate the performance of the four protocols they have proposed against some of the protocols like Epaox, Atlas and so on. Although the aim of this paper is not to evaluate performance of multi-leader protocols, graphs that illustrate the performance of the proposed variants can help understand various design choices, such as which scheme is better Tension Avoidance or Tension Resolution.
2. This paper needs some more work to clarify the different proposed protocols. Further, for several sections, there is a sudden switch from normal phase to recovery phase, which makes it difficult to understand. I explain most of my concerns next:

## Major comments:
1. While explaining the recovery for Simple PPaxos, the paper mentions "unchosen vertex" at the start of Section 4.4. It will be helpful if the paper explains what is meant by unchosen here. Further, it is important to re-iterate how can a vertex be unchosen.
2. I am unsure why at line 21 of Algorithm 1, the propose is sending to at least f+1 acceptors instead of all acceptors. As f of the acceptors may crash, so sending to all acceptors in important.
3. It is unclear to me what is the benefit of waiting for messages from all the 2f+1 acceptors in line 1, Algorithm 3. The paper should explain what will be the issue if the proposer only waits for messages from f + maj(f+1) acceptors. I understand that the change at line helps to resolve the safety bug of Fast PPaxos but not sure why change at line 1 important. Moreover, asking Unanimous Paxos's proposer to wait for messages from all the 2f+1 acceptors can hurt its liveness even if a single proposer fails. In such a case, how will the proposer recover the state. The paper needs to clarify this scenario.
4. I think there is a typo for lines 6-7 in Algorithm 3. They are not identical to the lines is Algorithm 1.
5. While explaining the Basic Epaxos optimization on Step 7, page 16, the paper states that "pi receives 2f votes for... including a vote from di". I am not sure when did di sent a vote to pi. In the presented algorithm, di only communicates with other dj. This step needs to be clarified.
6. It is unclear what is happening from steps (7) and (8) on Basic Epaxos. So, when pi receives 2f votes it sends v to ai. Then, ai records v as chosen and sends back to pi. Then pi sends to all other ancestors, and waits for acknowledgments. Why can't these steps be combined? Further, to reduce from 2f+1 to 2f, the Basic Epaxos optimization requires more network communication--from proposer to all acceptors and then acknowledgments from acceptors to the proposer. This needs to be clarified.

## Minor comments:
1. On Page 6, the paper states Consensus Invariants, which uses the terms (x, deps(x)). These terms have not been mentioned anytime before in this subsection. Although it can be understood by further reading, it will be helpful to the reader if explained before.
2. It will be helpful if the paper either uses the term replicas or learners. The paper switches between the terms, which makes it often confusing.
3. The following line "We consider what happens when not every..." on page 11, left hand side column are confusing the reader, as I expected that following this line the solution will be presented, but that happens much later in this section. Either consider removing this line or re-write it a little.
4. It is worth separating the recovery algorithm and not receiving enough matching votes (in Fast Paxos section) with suitable titles to improve readability.


**Expertise:**

Published in this area in the last 5 years

**Useful:**

yes

---

### Official Review · AnonReviewer1 · 2021-04-10
**A pedagogical paper on dependence graph-based consensus**

**Decision:**

Weak accept: good paper with flaws that can be fixed in three months

**Review:**

Summary

"SoK: A Generalized Multi-Leader State Machine Replication Tutorial" presents a pedagogical consensus protocol called PPaxos and several improvements on it. The main idea is that we can understand complex high-performance consensus protocols better by separating dependency service from the main Paxos protocol. It starts with Paxos and MultiPaxos, and then introduces the idea of conflict and dependency graph. By separating the dependency requirement, it presents two interesting concepts tension avoidance and tension resolution.

Strength
- Walking through the development of PPaxos, readers can build an in-depth understanding of dependency graph-based consensus protocols.

Weakness
- Although the paper targets to be a pedagogical paper, it is not easy to read. Cleaning up writing may be necessary.
- Paper often gets not precise. Clear assumptions and thorough correctness arguments are missing.

I think this paper has the potential to be a good tutorial to dependency graph-based consensus protocols. (I might change the title. I think this paper is specifically on dependency graph-based Paxos protocols, not for general multi-leader consensus protocols.) However, it wasn't a paper that reads smoothly, especially since this paper presents many different protocols. As it is targetting to be a pedagogical paper, I recommend authors to spend time on that. In the detailed comments below, I added a few suggestions.

Also, the paper gets loose on preciseness from time to time. I understand that it might be impossible to present proofs to all protocols presented. But it sometimes felt too loose to wrap my mind about correctness.

Finally, I felt the paper wanders among various PPaxos & other prior work. I wonder if setting up a goal upfront would be helpful. It might be useful to set the goal of understanding EPaxos like protocols and why people want them.

Page 1.
- In the introduction, I would clarify that this paper's scope is on dependency graph-based Paxos protocols. It

Page 2.
- "For example, with two proposers ... the leader of the round" seems not important.
- "chosen" at the end is not defined. It seems different from "chosen" in page 4. I would clarify "chosen" and "voted" in page 3.

Page 3.
- Figure 1 added little value to me. Except "number of leaders", terms in other boxes are not yet defined. Also, it covers many protocols not discussed in the paper.
- Ditto for Table 1. "maj()" is not defined either.
- I add details to Figure 3. The current version only shows how many message delays are required. I think a figure with terms (vr, vv, PHASE1A<i>, etc) used in the text will be very helpful.
- remove (see [15] for details)
- Meaning of "v" is mixed. Sometimes "v" means the value given by the client. Sometimes it's not.

Page 4.
- If any message gets ignored, what does a proposer do?
- Can you elaborate on how replicas work? Why is it safe, and how to ensure consistency?

Page 5.
- Need an explanation on directed edges. There is no introduction of the meaning of direction. The definition of dependency explained so far doesn't pose any order.
- 3.2, "Replicas execute conflict graphs in reverse topological order" => why? what goes wrong if executed in topological order? I could imagine it's like EPaxos, but unclear. I wish to see some explanation based on the ultimate goal.
- 3.3 "a log one entry" => typo?

Page 7.
- "v_x = (p_i, m)" => v_x, whose vertex id is (p_i, m). Mixing command, vertex, and vertex id was a bit confusing. For example, "globally unique vertex id v_x = (p_i, m)" but later in text, v_x means vertex, not id.

Page 8.
- Texts for figure 11 are difficult to read. Maybe move them to caption.
- Here, you introduce the timer. I think Paxos protocols generally don't require a synchronous network. I think this new requirement should be declared earlier.
- Separate client request (command) from vertex id. If not, it seems that clients won't be able to retry after the vertex gets noop.

Page 9.
- 4.5 doesn't prove that the two invariants guarantee linearizability or something similar. I would like to see the correctness argument for SMR, not just to the two invariants.
- "commit time" is typically defined as the time period between a client sends a request and the client knows it's committed. Execution is not a requirement. It's an important distinction for EPaxos, whose commit latency is different from execution latency (which may be longer than commit latency).
- I don't understand how EPaxos leverages FastPaxos. Also, EPaxos's key advantage is optimistic two wide-area message delays. More explanation is needed here on the delicate need of FastPaxos & why 4 message delays are considered "improvement" despite MultiPaxos also only needs 4 message delays. I guess adding multi-leader requires more message delays, which could be removed by "FastPaxos".

Page 10.
- Why use both learner and replica? Any difference?
- I feel "maj()" is awkward. Why not just use an equation? Also, I don't understand why maj(n) needs to be defined separately for odd n and even n.
- If you used the commit time defined earlier, it can get more than 4. It must wait for dependent commands.
- 6.2 Recovery => the term recovery is overloaded. Here, it's just adding noop. In other places, it commits with real command.
- In the bullet starting with "In Figure 14b," => remind readers about Paxos protocol. I think that's necessary to understand why "so is forced to propose (x, pi).
- I initially didn't understand why proposer p2 doesn't send the request to d1, d2, ... ,. It felt more natural to do that. But I guess you strictly abided by the Paxos protocol.

Page 14.
- 7.1 => first paragraph is complicated. Renaming phases and describing what they do would be helpful.

Page 15.
- Does Unanimous PPaxos modify the "Paxos" protocol part? Can you explain why replicas will still converge and agree?

**Expertise:**

Published in this area in the last 5 years

**Useful:**

yes

---

### Meta-Review · Area_Chair1 · 2021-04-14

**Recommendation:** Revise
**Confidence:** 5

**Metareview:**

Dear Authors, thank you for submitting to JSys. After reading the reviews, I recommend a REVISE decision. I believe the concerns of reviewers can be addressed in the next three months. After the submission of your revised version, reviewers will evaluate your modifications and a final decision will be made.

To summarize the reviews, all reviewers are mildly optimistic for a future acceptance of the paper. Isolating some major concerns:
- The verbiage of the paper could be improved considering the pedagogical focus of the work. Every reviewer has suggestions in that regard.
- Having a performance evaluation would greatly help the reader compare the solutions.
- A discussion about three aspects related to the consensus protocols, namely reconfiguration, catching up from the shared state, and single-leader vs multi-leader should be included to have a comprehensive and self contained comparison among solutions.

I would also like to suggest authors to include in the discussion a very recent paper published at Eurosys2021 named TEMPO. TEMPO (http://software.imdea.org/~gotsman/papers/tempo-eurosys21.pdf) is a leaderless consensus algorithm like EPaxos and Caesar.

We look forward to receiving your revised version.

Thank  you one more time for submitting,

Roberto Palmieri
Area co-chair of JSys

---

### Decision · Program_Chairs · 2021-04-14

Accept